# Testosterone exacerbates neutrophilia and cardiac injury in myocardial infarction via actions in bone marrow

Elin Svedlund Eriksson [1], Marta Lantero Rodriguez [1,20], Bente Halvorsen [2,3,20], Inger Johansson [1], Anna K. F. Mårtensson [1], Anna S. Wilhelmson [1,4,5], Camilla Huse [2,3,6], Thor Ueland [2,3,7], Pål Aukrust [2,3], Kaspar Broch [8,9], Lars Gullestad [3,8,9], Brage Høyem Amundsen [10,11], Geir Øystein Andersen [12], Mikael C. I. Karlsson [13], Malin Hagberg Thulin [14], Alessandro Camponeschi [15,16], Dana Trompet [14,17], Ola Hammarsten [18], Björn Redfors [1], Jan Borén [1], Elmir Omerovic [1], Malin C. Levin [1], Andrei S. Chagin [14], Tuva B. Dahl [2] & Åsa Tivesten [1,19] ✉

Men develop larger infarct sizes than women after a myocardial infarction (MI), but the mechanism underlying this sex difference is unknown. Here, we demonstrated that blood neutrophil counts post-MI were higher in male than female mice. Castration-induced testosterone deficiency reduced blood neutrophil counts to the level in females and increased survival post-MI. These effects were mimicked by Osterix-directed ablation of the androgen receptor in bone marrow (BM). Mechanistically, androgens downregulated the leukocyte retention factor CXCL12 in BM stromal cells. Post-hoc analysis of clinical trial data showed that neutrophilia was greater in men than women after reperfusion of first-time ST-elevation MI, and tocilizumab, an interleukin-6 receptor inhibitor, reduced blood neutrophil counts and infarct size to a greater extent in men than women. Our work reveals a previously unknown mechanism connecting testosterone with neutrophilia and MI injury via BM and identifies the importance of considering sex when developing anti-inflammatory strategies to treat MI.

An acute obstruction of the blood flow in the coronary arteries causes a myocardial infarction (MI), with potentially deleterious consequences. Studies have proposed that men have larger infarct sizes than women[1–3], even in relation to the area at risk[2], indicating that there may be a pathophysiological difference in infarct evolution between men and women. Further, when deaths outside hospital are taken into account, men appear to have a worse short-term prognosis after first-time MI[4]. Support for a male disadvantage in acute MI is also provided by preclinical studies showing that mortality and cardiac dilatation after an MI are higher in males compared with female mice[5–7]. Given that testosterone levels are more than ten-fold higher in men than in

women[8], testosterone-mediated effects may contribute to sex differences in MI injury. Indeed, testosterone deficiency induced by castration protects male rodents from post-MI injury[6,9] and testosterone treatment in female mice increases mortality and cardiac rupture post-MI[6].

The inflammatory response in the first days after an MI is an important determinant of infarct size and later clinical consequences[10,11]. The early phase post-MI is dominated by high levels of neutrophils, which contribute to myocardial injury[11]. Neutrophils mainly originate from the bone marrow (BM), and the release of neutrophils in acute MI is regulated by BM stromal cells (BMSCs)[12,13]. Both

cardiac infiltration of neutrophils and circulating neutrophil counts correlate positively with infarct size, ventricular arrhythmias, development of heart failure and mortality[14–16]. Several immunomodulatory strategies to reduce MI-reperfusion injury are currently being tested in clinical trials[17]. A promising example is the interleukin-6 (IL-6) receptor inhibitor tocilizumab, which has been shown to reduce both neutrophilia and myocardial damage in acute ST-elevation MI (STEMI)[18,19].

Sex differences in neutrophil trafficking have been reported[20,21], but it remains unclear whether there are sex differences in the neutrophilic response to an acute MI that impact cardiac injury. Further, castration of male mice reduces neutrophil levels in the heart in parallel with protection from post-MI injury[6], but whether there is a direct causal link between testosterone, neutrophil accumulation, and MI injury is unknown.

In this work, we show that testosterone exacerbates neutrophilia in experimental MI via actions in BMSCs and thereby worsens post-MI outcomes in males. We further show that men have more pronounced neutrophilia in acute MI than women and that anti-inflammatory treatment given in acute MI is more effective in men.

## Results

### Neutrophilia and cardiac injury in acute MI are greater in male than female mice

To determine whether there are sex differences in neutrophilia in acute MI in mice, we compared male and female mice after experimental MI-reperfusion (induced by ligation of the left coronary artery for 45 min followed by 24 h reperfusion). We showed that blood neutrophil count at 24 h was indeed higher in male mice (Fig. 1a). Further, despite similar areas at risk in male and female mice, male mice developed larger infarct sizes (Fig. 1b). Using a high-sensitive assay, we confirmed that circulating testosterone levels were higher (15-fold) in male versus female mice at 24 h after reperfusion (Supplementary Fig. 1a). We also showed that the MI surgery per se did not affect endogenous testosterone levels in male mice (Supplementary Fig. 1b).

### Castration of male mice reduces neutrophilia, cardiac injury and cardiac remodeling and increases survival after an acute MI

We next compared castrated and sham-castrated male mice that similarly underwent MI-reperfusion surgery. We confirmed that castration of male mice reduced testosterone levels (−91%; Supplementary Fig. 1c) and showed that the blood neutrophil count in castrated male mice was similar to the level observed in female mice after 24 h reperfusion (Fig. 1c). Despite similar area at risk in castrated and sham-castrated male mice, castrated male mice developed smaller infarct sizes (Fig. 1d). Thus, testosterone potentially contributes to the higher level of blood neutrophil counts and cardiac injury observed in male mice in MI-reperfusion.

By comparing the accumulation of leukocytes in the heart of castrated and sham-castrated male mice after an experimental MI (induced by permanent ligation of the left coronary artery), we showed that castration reduced the number of neutrophils (Fig. 1e and Supplementary Fig. 2) but did not affect the number of monocytes or macrophages (Fig. 1f, g) in the left ventricle (LV) 48 h post-MI. Area at risk, estimated by echocardiography 24 h post-MI, was similar in castrated and control mice (Fig. 1h), but survival during a 3-week follow-up was greatly increased in castrated mice (Fig. 1i). The increased survival was explained by a greatly reduced cardiac rupture rate (Fig. 1j). Echocardiography of surviving mice at 3 weeks post-MI showed no differences in cardiac function (Supplementary Fig. 3a–d). However, 3 weeks after MI, LV volume in diastole was lower in the castrated group (Fig. 1k, l).

To confirm the adverse effects of testosterone, we administered a vehicle or a physiological dose of testosterone to castrated male mice. Plasma levels of the neutrophil-released enzyme myeloperoxidase (Fig. 1m) and the cardiac injury marker troponin I (Fig. 1n) were higher

in testosterone-treated versus vehicle-treated mice 24 h after permanent ligation of the left coronary artery.

Collectively, these results show that testosterone worsens cardiac injury in acute MI and that castration of male mice protects against neutrophilia, cardiac neutrophil infiltration, cardiac injury, mortality, and cardiac dilatation after an MI.

### Osterix-directed knockout of the AR reduces neutrophilia and cardiac remodeling and increases survival after an acute MI

We next addressed the potential mechanism(s) linking testosterone with neutrophilia in acute MI. In acute inflammation, neutrophils are mainly released from the BM into the blood, a process regulated by BMSCs[13]. To assess if testosterone affects neutrophil release through actions on BMSCs, we knocked out the androgen receptor (AR), the receptor for testosterone, in BMSCs. To achieve this, we used Osterix (Osx)-Cre, a classical Cre line that targets bone osteoprogenitors and BMSCs[22–24], and crossed Osx-Cre with AR-floxed mice, naming the resulting model O-ARKO[25]. As expected, Osx-targeted cells found in BM morphologically resembled BMSCs[26] as well as osteolineage cells (Fig. 2a). Analysis of the mouse model also revealed reporter-positive cells in the splenic stroma and sporadically in the heart (Supplementary Fig. 4a). However, Ar DNA content, determined as the ratio between floxed exon 2 and intact exon 3, was reduced by 50% in the femur shaft but unaltered in the spleen and heart (Supplementary Fig. 4b), suggesting low efficiency of AR ablation in these tissues.

Using the O-ARKO model, we tested the hypothesis that Osx-directed knockout of the AR in male mice has the same effect as castration on neutrophilia in acute MI. Indeed, we observed that blood neutrophil counts in experimental MI-reperfusion (45 min ischemia followed by 24 h reperfusion) were lower in O-ARKO mice than in control mice (Fig. 2b and Supplementary Fig. 5a). Further, despite similar areas at risk in O-ARKO and control mice, O-ARKO mice developed smaller infarct sizes (Fig. 2c).

After 24 h of reperfusion, blood neutrophils from O-ARKO and control mice showed similar Ly6G mean fluorescence intensity (Supplementary Fig. 5b), indicating similar maturity of circulating neutrophils[27]. Further, blood monocyte or lymphocyte levels did not differ between O-ARKO and control mice in experimental MI-reperfusion (Supplementary Fig. 5c–e).

Quantification of leukocyte subpopulations in the LV 48 h after an experimental MI by permanent ligation of the left coronary artery showed that the number of neutrophils was clearly lower in O-ARKO mice than in control mice (Fig. 2d) but the number of monocytes and macrophages was similar between the two groups of mice (Fig. 2e, f). The area at risk 24 h after MI was similar between O-ARKO and control mice (Fig. 2g), but O-ARKO mice showed a dramatically increased survival in the first 3 weeks post-MI (Fig. 2h) and complete protection from cardiac rupture (Fig. 2i), again mirroring the effect of castration. Three weeks after an MI, no difference in cardiac function (Supplementary Fig. 6a–d) was detected among surviving mice from the two groups. However, in line with the observations in castrated mice, 3 weeks after MI, LV volume in diastole was reduced in the O-ARKO mice (Fig. 2j, k).

These observations mirror the effects of castration and indicate that regulation of circulating/infiltrating neutrophils and exacerbation of post-MI injury by testosterone are mediated via AR-dependent effects on BMSCs.

### Cytokine-controlled neutrophilia depends on the AR in BM stroma

We next addressed the functional regulation of neutrophil release by the AR in BM stroma. To this end, we tested the effect of injecting O-ARKO male mice with granulocyte colony-stimulating factor (G-CSF), a potent mobilizer of neutrophils from the BM[13,28] that has been implicated in their egress in acute MI[13]. In the steady state (i.e., before

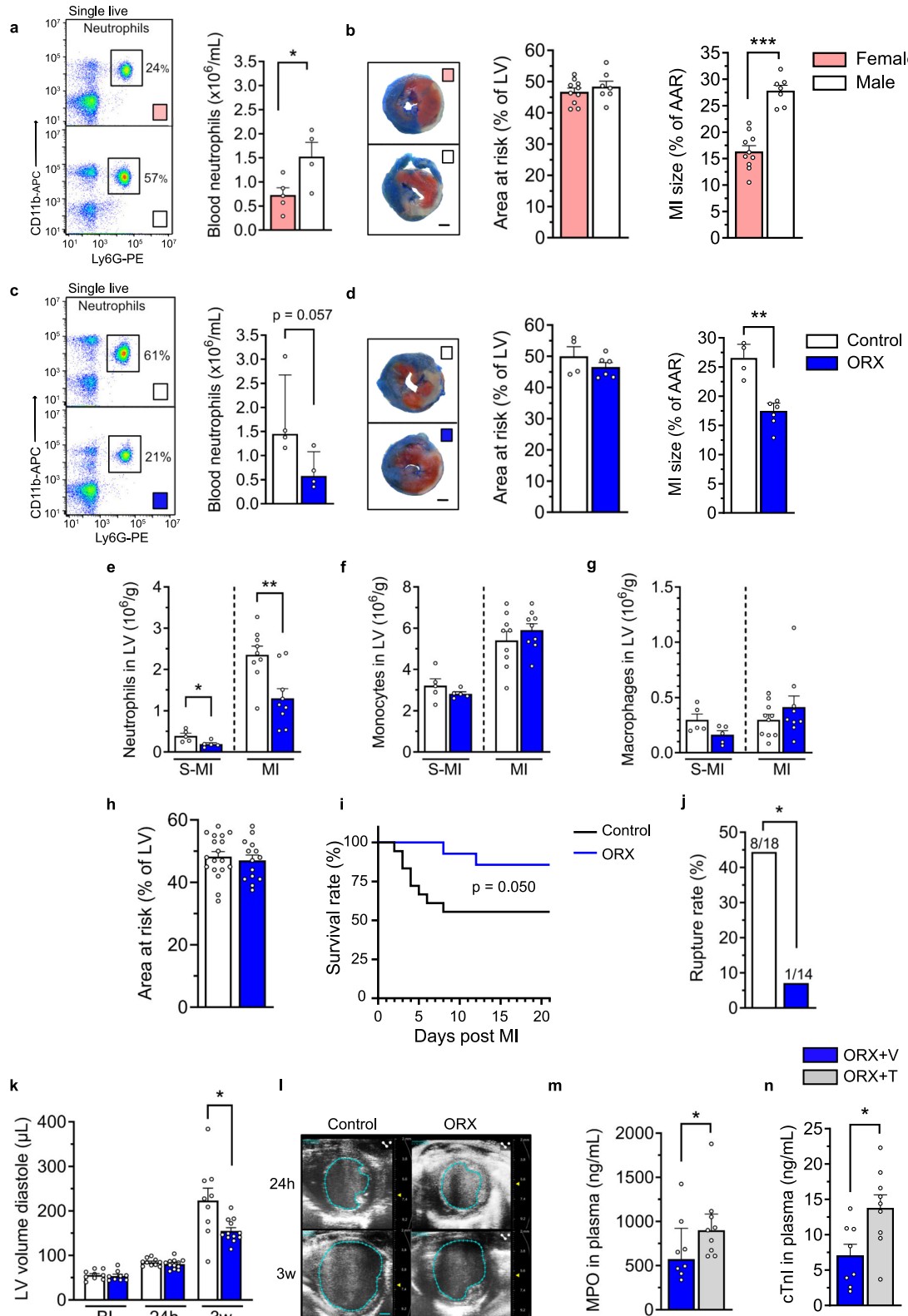

G-CSF injection), BM granulopoiesis (Supplementary Fig. 7a) and numbers of common myeloid progenitors[29] in BM (Supplementary Fig. 7b, c) and neutrophils in bone, blood, and spleen (Supplementary Fig. 7d–h) did not differ between O-ARKO and control mice. Further, apart from a small reduction in the number of blood platelets in O-ARKO mice, other major cell populations (lymphocytes, monocytes, eosinophils, erythrocytes) in the blood (Supplementary Fig. 7i–n) were

not different between O-ARKO and control mice. However, after G-CSF injection, we observed that neutrophilia was lower in O-ARKO versus control mice (Fig. 3a). Similarly, the extent of G-CSF-induced neutrophilia was lower in castrated than in non-castrated male mice (Fig. 3b).

The retention factor CXCL12, which is produced by BMSCs[22,30], is an important regulator of leukocyte retention and egress from the

**Fig. 1 | Castration of male mice reduces neutrophilia, cardiac injury, and cardiac remodeling and increases survival after an acute MI. a–d** Neutrophil counts in blood, area at risk (AAR) and MI size after 45 min ischemia followed by 24 h reperfusion. **a, b** Male and female mice; **a** *P = 0.038 (two-sided unpaired Student t test), n = 5 + 4; **b** ***P < 0.0001 (two-sided unpaired Student t test), n = 10 + 7. **c, d** Sham-castrated (control) and castrated (orchiectomized; ORX) male mice; **c** P = 0.057, n = 4 + 4 (two-sided Mann–Whitney); **d** **P = 0.0095 (two-sided Mann–Whitney), n = 4 + 6. **e–g** MI was induced by permanent ligation of the left coronary artery in ORX or sham-castrated (control) mice. 48 h after MI or sham-MI (S-MI) surgery, neutrophils (**e**), monocytes (**f**), and macrophages (**g**) were quantified in the left ventricle (LV) by flow cytometry. **e** *P = 0.017 and **P = 0.004 (two-sided unpaired Student t test). **e–g** n = 5 + 5 and 9 + 9 for S-MI and MI, respectively. **h–l** MI was induced by permanent ligation of the left coronary artery in ORX or sham-castrated (control) mice. Echocardiography was performed before MI (baseline;

BL) and 24 h and 3 weeks post-MI. **h** The area at risk analyzed at 24 h post-MI. **i** Survival rate; P = 0.050 (log-rank test). **j** Cardiac rupture rate during the 3 weeks of follow-up; *P = 0.044 (two-sided Fisher´s test); **h–j** n = 18 + 14. LV volume in diastole at BL, 24 h and 3 weeks post-MI; n = 9 + 9 (BL), 10 + 12 (24 h), and 9 + 12 (3w), 3w, *P = 0.042 (two-sided unpaired Welch´s t test). **l** Representative echocardiographic images in sham-operated (control) and castrated (ORX) mice at 24 h and 3 weeks post-MI, scale bar = 1 mm. **m, n** Male mice were castrated and treated with vehicle (V) or testosterone (T) for 3 weeks before permanent ligation of the left coronary artery. 24 h after MI, plasma was assessed for myeloperoxidase (MPO), **m** *P = 0.050 (two-sided unpaired Student t test) and cardiac troponin I (cTnI), **n**; *P = 0.015 (two-sided unpaired Student t test); **m, n**; n = 8 + 9. Bars indicate means (**a, b, e–g, k, m, n**) or medians (**c, d**), error bars are SEM (**a, b, e–g, k, m, n**) or interquartile ranges (**c, d**), and circles represent individual mice. Source data are provided as Source data file.

BM[31]. A reduction in BM CXCL12 has been implicated in the egress of BM neutrophils in acute MI[13] and a homeostatic increase in BM CXCL12 may theoretically be associated with reduced neutrophil egress[30]. Therefore, we next explored the potential AR-mediated regulation of this cytokine. Consistent with our hypothesis, levels of *Cxcl12* mRNA in non-hematopoietic (CD45⁻) BM cells were almost twice as high in cells from O-ARKO and castrated male mice than in cells from their respective controls (Fig. 3c, d). In sorted BMSCs from Osx-Cre tdTomato reporter mice, *Cxcl12* RNA levels were 7 times higher in tomato⁺ versus tomato⁻ cells (Fig. 3e, Supplementary Fig. 8), confirming that the Osx-Cre preferentially targets *Cxcl12*-expressing BMSCs.

To investigate the distribution of *Ar* and *Cxcl12* mRNA within bone, we used a previously published single-cell RNA sequencing dataset of mouse stromal cells[32]. We showed that both *Cxcl12* and *Ar* were expressed in mesenchymal stem/stromal cell and osteolineage cell clusters (Fig. 3f–k).

Finally, we incubated BMSCs[23] isolated from male C57BL/6 J mice with the AR agonist dihydrotestosterone. The AR agonist reduced *Cxcl12* mRNA expression in BMSCs (Fig. 3l) and CXCL12 protein levels in culture medium (Fig. 3m).

Taken together, these findings indicate that androgens downregulate the expression of the leukocyte retention factor CXCL12 in BMSCs via the AR. Lower BM levels of this cytokine may underlie the higher release of neutrophils in males.

### Neutrophilia, cardiac injury, and response to anti-inflammatory treatment in acute MI are greater in men than women

We next tested the hypothesis that neutrophilia in acute MI is more pronounced in men than women and that anti-inflammatory treatment given in acute MI is consequently more effective in men. To this end, we performed a post-hoc analysis of data from the randomized ASSAIL-MI (ASSessing the effect of Anti-IL-6 treatment in MI) trial, in which men and women with first-time STEMI received a single dose of placebo or tocilizumab prior to reperfusion[18] (Fig. 4a). Previous analyses of a broad spectrum of leukocyte subpopulations in ASSAIL-MI showed that the blood neutrophil count was significantly lower in the tocilizumab group (men and women combined)[19].

To explore sex differences, we stratified ASSAIL-MI trial participants by treatment allocation and sex (baseline characteristics in Supplementary Table 1). In the placebo group, blood neutrophil counts were higher in men than women at the first two time points after reperfusion (24 h and 3–7 days) but were similar in both sexes at later time points (3 and 6 months) (Fig. 4b and Supplementary Fig. 9a). Tocilizumab dramatically reduced blood neutrophil counts at 24 h and 3–7 days. However, the effect of tocilizumab was greater in men, as tocilizumab treatment reduced blood neutrophil counts at these time points to similar levels in men and women, and there was a significant interaction between sex and treatment on circulating neutrophils (Fig. 4b).

In contrast to neutrophils, there were no statistically significant interactions between sex and treatment for other cell types examined, including monocytes, lymphocytes, and platelets (Supplementary Fig. 9b–d). Troponin T levels (analyzed at admission and 8 h, 16 h, 24 h, 3–7 days and 3 and 6 months after reperfusion) showed no significant interaction between sex and treatment; however, tocilizumab reduced troponin T levels in men at 16 h after reperfusion (Supplementary Fig. 10).

Cardiac magnetic resonance imaging performed 3–7 days after reperfusion showed that the infarct size adjusted for LV mass was larger in men than women in the placebo group (Fig. 4c, d and Supplementary Fig. 11a, b). Tocilizumab significantly reduced infarct size at 3–7 days in men, and there was a statistical interaction between sex and treatment on early infarct size (Fig. 4d). Cardiac magnetic resonance imaging performed 6 months after reperfusion showed similar results (Fig. 4e and Supplementary Fig. 11c).

In summary, these results show that men with first-time STEMI display more pronounced neutrophilia compared to women. Administration of the IL-6 receptor inhibitor tocilizumab prior to reperfusion reduced blood neutrophil counts and reduced MI size more markedly in men than in women, demonstrating important interactions between sex and anti-inflammatory therapy in acute MI.

## Discussion

In this study, we showed that testosterone via the AR exacerbates the egress of neutrophils from the BM in acute MI and worsens post-MI outcomes in males. In experimental MI-reperfusion, blood neutrophil counts were higher in male versus female mice. In male mice, both castration-induced testosterone deficiency and Osx-driven depletion of the AR in bone cells, including BMSCs, reduced neutrophilia and neutrophil accumulation in the heart and protected against cardiac injury and dilatation and mortality post-MI. G-CSF-stimulated neutrophilia was also lower in male castrated and O-ARKO mice compared with their respective controls. Further, we showed that CXCL12, a pivotal leukocyte retention factor, was downregulated by AR activation in BMSCs, in support of a direct influence of androgens on neutrophil egress from the BM. The relevance of these findings to humans was shown in an analysis of clinical trial data, in which we observed that neutrophilia after a first-time STEMI was more pronounced in men than women. Furthermore, treatment with tocilizumab, an IL-6 receptor inhibitor, prior to reperfusion reduced blood neutrophil counts and infarct size to a greater extent in men than women, demonstrating interactions between sex and anti-inflammatory therapy in acute MI.

A central finding of the present study is that males with acute MI showed more pronounced neutrophilia and increased infarct size compared to females in both the clinical and experimental settings. This finding is consistent with the higher numbers of neutrophils in the infarct border zone and larger post-MI injury in male versus female

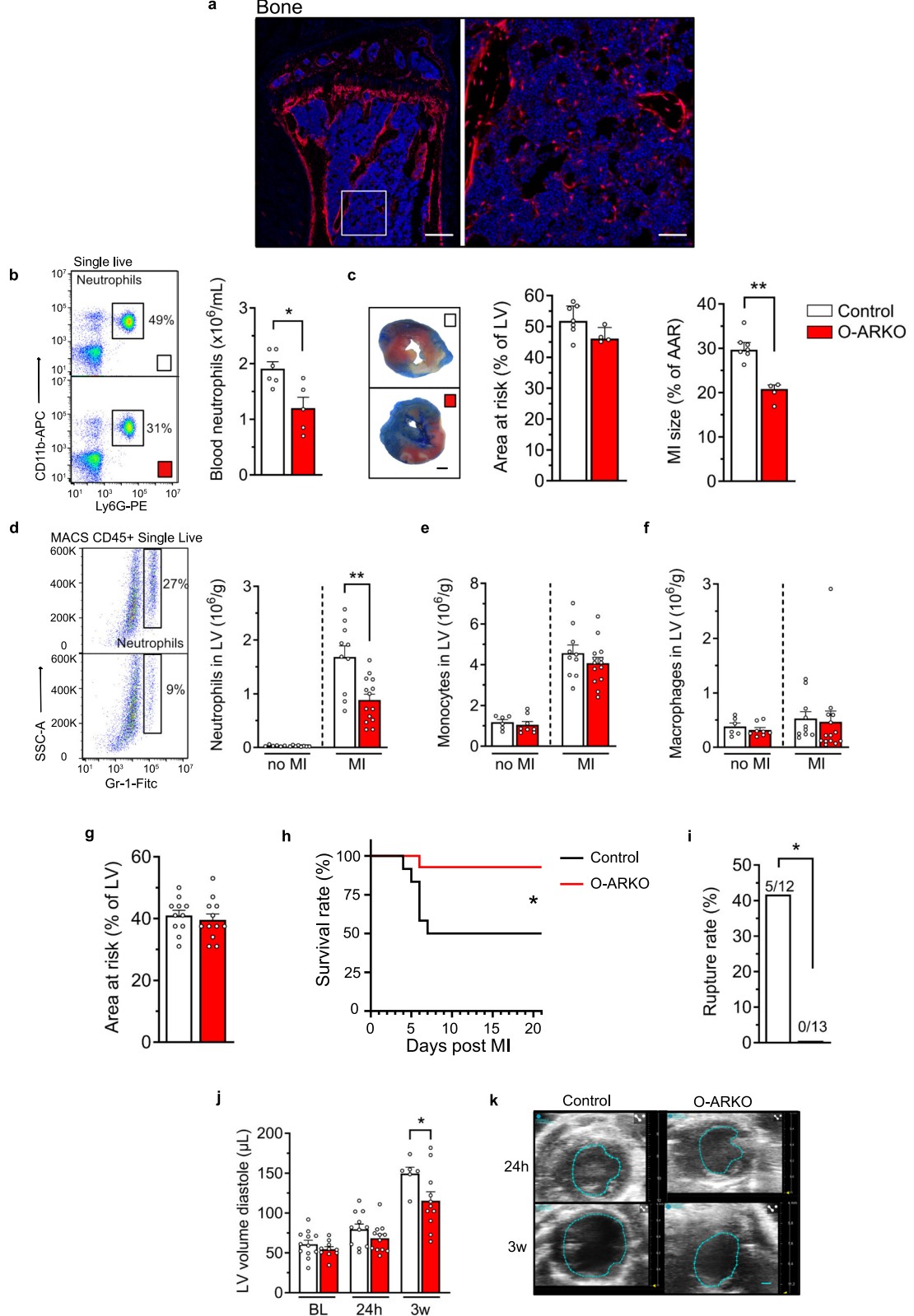

mice observed in earlier studies[5,33]. Sexual dimorphism in neutrophilia in acute inflammatory states is also supported by previous studies. For example, in models of mesenteric ischemia-reperfusion and sterile peritonitis, male mice show a higher degree of neutrophil mobilization than females[34,35]. Furthermore, sex-specific effects on neutrophil trafficking have been suggested to underlie sex differences in outcomes

after infections[20,21]. We further demonstrated here that castration of male mice reduced the neutrophilia to the levels of female mice in acute MI. Strikingly, this effect of castration was mimicked by knockout of the AR in BMSCs. Both castration and knockout of the AR in BMSCs of male mice reduced cardiac neutrophil accumulation, infarct size, mortality, and cardiac dilatation post-MI. Collectively, these data

**Fig. 2 | Osterix-directed knockout of the androgen receptor reduces neutrophilia and cardiac remodeling and increases survival after an acute MI. a** A representative image of Osterix-Cre-directed tdTomato reporter signal in femur. Scale bars = 250 and 50 μm (zoomed image), respectively. **b, c** Neutrophil counts in blood, area at risk (AAR), and MI size after 45 min ischemia followed by 24 h reperfusion in O-ARKO and control (Osx1-Cre⁺) mice. **b** *P = 0.012 (two-sided unpaired Student t test); n = 6 + 5. **c** **P = 0.0061 (two-sided Mann–Whitney); n = 7 + 4. **d–f** MI was induced by permanent ligation of the left coronary artery in control and O-ARKO mice. Mice without MI surgery were used for reference. 48 h after the MI surgery, neutrophils (**d**), monocytes (**e**), and macrophages (**f**) were quantified in the left ventricle (LV) by flow cytometry. **d** **P = 0.0013 (two-sided unpaired Student t test), **d–f** n = 6 + 8 and 10 + 14 for no MI and MI, respectively.

**g–k** MI was induced by permanent ligation of the left coronary artery in O-ARKO or control mice. Echocardiography was performed before MI surgery (baseline; BL) and 24 h and 3 weeks post-MI. **g** The area at risk analyzed at 24 h post-MI; n = 11 + 12. **h** Survival rate; *P = 0.014 (log-rank test). **i** Cardiac rupture rate during the 3 weeks of follow-up; *P = 0.015 (two-sided Fisher´s test); **h–j** n = 12 + 13. **j** LV volume in diastole at BL, 24 h and 3 weeks post-MI; n = 12 + 10 (BL), 11 + 12 (24 h), 6 + 11 (3w). 3w, *P = 0.048 (two-sided Mann–Whitney). **k** Representative echocardiographic images in sham-operated (control) and castrated (ORX) mice at 24 h and 3 weeks post-MI. Scale bar = 1 mm. Bars indicate means (**b**, **d–g**, **j**) or medians (**c**), error bars are SEM (**b**, **d–g**, **j**) or interquartile ranges (**c**) and circles represent individual mice. Source data are provided as Source data file.

suggest that testosterone, via the AR in BMSCs, increases neutrophil egress from the BM in acute MI, leading to increased myocardial damage (Fig. 5).

In acute MI, the initial increase in blood neutrophils is likely a result of a mass exodus of neutrophils from the BM, facilitated by raised levels of inflammatory factors such as G-CSF and IL-6 in the blood and suppression of neutrophil retention factors, most importantly CXCL12, in the BM[13]. We showed that AR activation reduced levels of both *Cxcl12* mRNA and CXCL12 protein in BMSCs. We thus propose that androgen/AR depletion alters the function of BMSCs such that they constitutively express higher levels of CXCL12, and that the BM is thereby less responsive to triggers of leukocyte egress, such as G-CSF[30] (Fig. 5). Consistent with this proposal, both O-ARKO and castrated mice displayed reduced G-CSF-induced neutrophilia.

BMSCs regulate both neutrophil egress and hematopoiesis[12,22]. In the steady state, testosterone-deficient whole-body AR knockout or castrated male mice show a defective granulopoiesis from the later myelocyte stage and a reduced number of neutrophils in blood[25,36–38]. In accordance, testosterone treatment of men dose-dependently increases blood neutrophil counts[39]. By contrast, we showed that steady-state granulopoiesis and blood neutrophils were unaltered in O-ARKO mice. Our observation that neutrophilia after MI-reperfusion was lower both in castrated male mice and O-ARKO male mice compared with their respective controls suggests the importance of androgen/AR-mediated regulation of neutrophil egress, rather than granulopoiesis, in the acute MI setting.

In addition to its effects on neutrophils, testosterone therapy dose-dependently increases the circulating counts of erythrocytes, monocytes, and platelets in men[39] and androgen deprivation modestly lowers erythrocyte counts in men[40]. These observations raise the question of whether androgen affects early myeloid progenitors in the BM. However, circulating counts of monocytes, platelets, and erythrocytes and numbers of myeloid progenitors are not altered by whole-body AR knockout in male mice[36]. Similarly, in our current study, O-ARKO male mice displayed no alterations in circulating counts of monocytes or erythrocytes, although blood platelet numbers were slightly reduced. Further, we did not detect any differences in the number of common myeloid progenitors in the BM of O-ARKO compared with control mice.

CXCL12 production from Osx-Cre-targeted stroma is required for the efficient retention of leukocytes in the BM, but also for the maintenance of B-lymphoid-committed progenitors[22]. We have previously shown that both whole-body AR knockout and O-ARKO male mice display increased BM B lymphopoiesis[25]. Intriguingly, it was recently shown that BM B cells themselves are important for the retention/egress of leukocytes. B cells produce acetylcholine, which acts on BMSCs to regulate their expression of *Cxcl12*, and depletion of the acetylcholine-producing enzyme in B cells aggravates post-MI injury in mice[41]. Thus, although we demonstrate that androgens/AR regulate CXCL12 in BMSCs in a B cell-independent manner, it is possible that BM-resident B cells also contribute to the regulation of neutrophilia in the in vivo setting.

In accordance with the preclinical observations, we showed that MI-reperfusion-induced neutrophilia was more pronounced in men than women. Further, infarct size was greater in men than women in ASSAIL-MI. As the initial neutrophil response contributes to the myocardial injury in acute MI[11], we propose that the sex differences in infarct size in ASSAIL-MI trial participants may be at least partly explained by sex differences in neutrophilia. This notion is supported by the sexually dimorphic response to the IL-6 receptor inhibitor tocilizumab with respect to both neutrophilia and MI size in the ASSAIL-MI trial.

After MI, both circulating and myocardial levels of IL-6 increase, and cardiac fibroblasts are a major source of IL-6 following MI[42]. A central role for IL-6 as a trigger of neutrophilia in human MI is supported by the strikingly lower blood neutrophil levels in the tocilizumab-treated group in ASSAIL-MI (Fig. 5). In line with this notion, IL-6 has been shown to mobilize neutrophils from the BM into the circulating pool[43], adding effects on BM neutrophil release to the list of potential cardioprotective actions of IL-6 receptor inhibition[18,44]. While a reduction in BM CXCL12 has been implicated in the egress of BM neutrophils in acute MI[13], a potential role of CXCL12 in the neutrophilic response specific to IL-6 remains unclear.

Testosterone levels are more than 10-fold higher in men than in women[8]. Using a high-sensitivity assay, we showed that serum testosterone levels were 15-fold higher in male than female mice and observed an expected natural variation in testosterone levels between male mice[45]. We found no effect of MI (versus sham-MI) on testosterone levels in male mice 48 h after surgery. We did not measure testosterone levels in ASSAIL-MI and, therefore, cannot determine how testosterone levels were potentially affected by MI or tocilizumab in this trial. Testosterone levels in men have been reported to be lowered after acute MI (within 11 h)[46,47], although the reported reduction was small (-20%)[47]. Indeed, the complex interplay between acute/chronic inflammatory and stress conditions and anabolic hormones[48], including testosterone, may be highly relevant for the acute MI setting.

The clinical data on the cardiovascular actions of androgens are conflicting, and there is a lack of understanding of underlying mechanisms. Some data link low testosterone levels and/or androgen deprivation to increased cardiovascular risk in men[49,50]. Conversely, some evidence supports the adverse effects of testosterone, including the positive association of genetically determined testosterone levels with the risk of MI and heart failure in men[51], and the increased cardiovascular risk in transmen on testosterone treatment[52]. Notably, a large trial of testosterone replacement therapy in men with testosterone deficiency found no significant differences in major adverse cardiac events, MI, or death due to cardiovascular causes between testosterone- and placebo-treated men[53] and testosterone treatment for 3 years was not associated with acceleration of atherogenesis[54]. Among experimental studies, several report that castration/antiandrogen treatment of rodents protects against the adverse consequences of experimental MI[5,6,9,55–57], while others show no beneficial effects of testosterone in experimental MI models[58–62]. A potential explanation for these diverging results may be that different

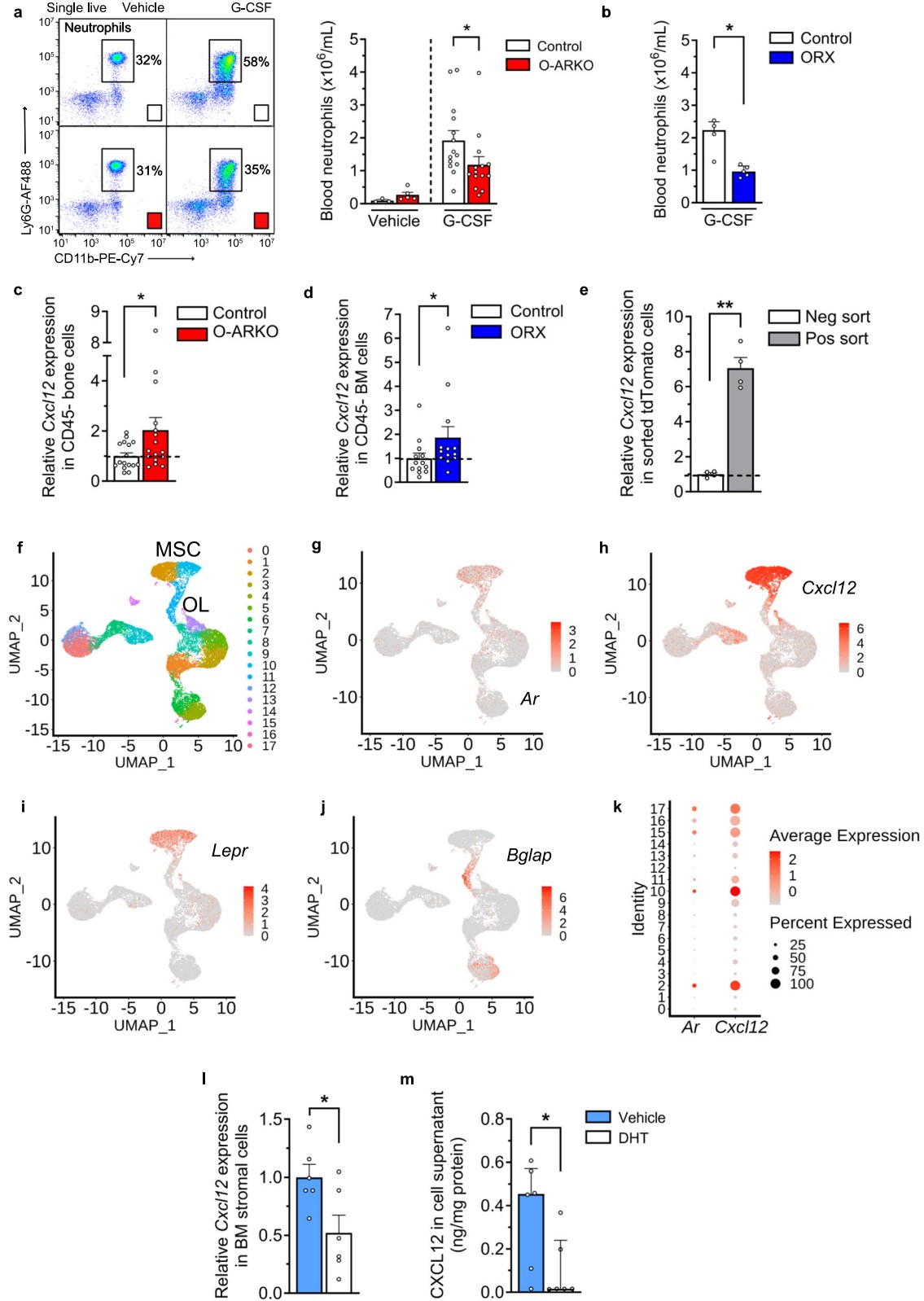

mechanisms are in play, including (anti)atherogenic[63], metabolic, direct cardiotropic[64] and immunomodulatory actions[63,65], and that their relative contribution varies depending on experimental settings, target tissues, cardiovascular diseases and disease stages.

Our results have important implications. First, they provide a potential explanation for sex differences in acute MI size and complications reported in both humans and mice[1–7] and show that men and women may respond differently to anti-inflammatory therapies in acute MI. Of note, clinical trials in this area are rarely designed to determine sex-specific effects[66]. In the era of precision medicine, sex differences must be considered when developing anti-inflammatory strategies to reduce MI injury.

Second, our results raise the issue of whether drugs targeting androgen levels/action affect clinical consequences after an MI. This

**Fig. 3 | Cytokine-controlled neutrophilia depends on the AR in BM stroma.**
**a, b** Mice were injected i.p. twice daily with 125 μg/kg of G-CSF or vehicle on three consecutive days. Neutrophil (Ly6G$^+$) counts in blood were measured by flow cytometry; representative flow plots in **a**. **a** Vehicle or G-CSF injection in control or O-ARKO mice; *$P = 0.034$ (two-sided unpaired Student $t$ test); $n = 3 + 5$ and $14 + 14$ for vehicle and G-CSF respectively. **b** G-CSF injection in sham-castrated (control) or castrated (ORX) mice; *$P = 0.016$ (two-sided Mann–Whitney); $n = 4 + 5$. **c, d** *Cxcl12* expression was analyzed by qPCR in CD45$^-$ (CD45/TER119 depleted) bone cells from O-ARKO or control mice (**c**) and CD45$^-$ (CD45$^+$ depleted) bone marrow (BM) cells from ORX or control mice (**d**). **c** *$P = 0.031$ (two-sided unpaired Student $t$ test); $n = 16 + 16$. **d** *$P = 0.037$ (two-sided unpaired Student $t$ test); $n = 13 + 13$. **e** BM cells from Osterix-Cre$^+$ tdTomato reporter mice were cultured for 10 days, tdTomato+ and tdTomato- cells were sorted and *Cxcl12* expression analyzed by qPCR; **$P = 0.0018$ (two-sided unpaired Welch's $t$ test); $n = 4 + 4$. **f–k** Analysis of *Ar* and

*Cxcl12* distribution in a single cell RNA sequencing dataset of mouse stromal cells (ref. 32). **f** In the overview of 18 stromal cell clusters colored by clustering (UMAP visualization), clusters 2 and 10 are multipotent mesenchymal stem/stromal cells (MSC) and cluster 11 osteolineage (OL) cells. **g–j** The expression intensity and distribution of *Ar*, *Cxcl12*, *Lepr*, and *Bglap* RNA in stromal cell clusters. **k** Percentages and expression intensity of *Ar* and *Cxcl12* in stromal cell clusters. **l, m** BM stromal cells from C57BL/6 J male mice were stimulated with the AR agonist dihydrotestosterone (DHT) or vehicle for 10 days. **l** *Cxcl12* mRNA expression; *$P = 0.028$ (two-sided unpaired Student $t$ test); $n = 6 + 6$. **m** CXCL12 protein levels in the cell supernatant; *$P = 0.048$ (two-sided Mann–Whitney); $n = 6 + 6$. Bars indicate means (**a, c–e, l**) or medians (**b, m**), error bars are SEM (**a, c–e, l**) or interquartile ranges (**b, m**), and circles represent individual mice. Source data are provided as Source data file.

has potential clinical relevance in large patient groups that are already treated with androgen/AR-modulating drugs (e.g., for prostate cancer, prostate hyperplasia, hypogonadism or gender dysphoria). Disentangling mechanisms and target organs will be crucial for insights into the cardiovascular profile of androgen-modulating drugs, and an increased understanding of AR target cells, as provided here, may have important implications for the development of selective AR modulators (SARMs). While considerably less developed than selective estrogen receptor modulators that are in clinical use, SARMs show therapeutic promise and have been applied to several diseases[67].

Our study has limitations. In response to acute stress such as MI, hematopoietic cells may egress from the spleen besides the BM[68]. In our model, we found tomato-positive cells in the splenic stroma, in line with previous data that Osx-driven Cre expression is not entirely specific to bone[69]. However, we did not detect measurable levels of *Ar* DNA recombination in the spleen, likely reflecting the low efficiency of Osx-driven Cre in splenic cells. Further, while the spleen plays a significant role in the inflammatory response to an MI[68], experimental evidence supports the role of the BM over the spleen as a supplier of neutrophils in the early stages of an MI[13,68]. Nevertheless, we cannot exclude a contribution of the AR in splenic stromal cells for the O-ARKO data reported here. Neutrophils may mediate both adverse and beneficial effects post-MI[21] and whether androgens confer any beneficial effects on the long-term healing and revascularization after MI remains to be addressed. While castration and O-ARKO effects in male mice were studied in both reperfused and non-reperfused MI, the effect of testosterone replacement was studied in the non-reperfused MI model only. We studied mice at 3–5 months of age when they were homogenous in physiological maturation and body weight[70], but we did not determine effects in older mice. Further, we did not study the role of endogenous androgens in females, which would require alternative experimental approaches given the location of the AR on the X chromosome and the concomitant loss of androgens and estrogens following castration/ovariectomy of females. Another limitation is the low number of women in the ASSAIL-MI trial and the high proportion of white participants, which limits the generalizability to other racial/ethnic groups. The analyses in ASSAIL-MI trial participants performed in this study were exploratory (i.e., post-hoc). The results, therefore, need confirmation in trials addressing sex differences and sex-specific effects in the response to anti-inflammatory therapy in acute MI.

In conclusion, we have shown that testosterone, via the AR in BMSCs, exacerbates the egress of neutrophils from the BM in acute MI and worsens post-MI injury in males. In accordance, we also showed that (1) neutrophilia is greater in men than women and (2) anti-inflammatory treatment reduces blood neutrophil counts and infarct size to a greater extent in men than women. This work reveals a previously unknown mechanism connecting testosterone with neutrophilia and MI injury via bone and identifies the importance of

considering sex when developing anti-inflammatory strategies to treat MI.

## Methods

### Animals
Mice on C57BL/6 J background (Jackson Laboratory, Bar Harbor, ME, USA; JAX000664) were used in all mouse experiments. Male mice with Osterix (Osx1)-Cre-mediated inactivation of the AR (O-ARKO mice) were generated by breeding AR (AR)$^{+/flox}$ female mice[71] with male Osx1-Cre$^+$ mice (Jackson Laboratory, Bar Harbor, Maine, USA; JAX006361), as previously described[25]. Due to congenital malocclusion in Osx1-Cre$^+$ mice[72], Osx1-Cre$^+$ controls, and O-ARKO mice were fed soaked chow up to 8 weeks of age, and teeth were cut to maintain chewing ability, resulting in normalized growth. Osx1-Cre$^+$ littermate mice were used as controls in all O-ARKO experiments. Male tdTomato$^+$ Osx1-Cre$^+$ mice were generated by breeding homozygous tdTomato$^{+/+}$ mice (Rosa26-CAG-loxP-stop-loxP-tdTomato; Jackson Laboratory, Bar Harbor, Maine, USA; JAX007914)[73] with Osx1-Cre$^+$ mice. The presence of AR$^{flox}$, Cre and tdTomato was assessed by PCR amplification of genomic DNA. If not otherwise specified, mice were studied at 3–5 months of age. The mice were housed in a temperature- and humidity–controlled room with a 7:00-19:00 h light cycle and consumed a soy-free diet (RM3(E)-soy-free, SDS; Teklad Global 16% Protein Rodent diet 2016, Harlan Laboratories or R70, Lantmännen) and tap water ad libitum. The Ethics Committee on Animal Care and Use in Gothenburg approved all procedures.

### Castration
The mice were anesthetized with 2.5% isoflurane (IsoFlo® vet., Vnr 002185, Zoetis) and bilaterally castrated (ORX) or sham-ORX at 9–15 weeks of age. Buprenorphine (Temgesic®, RB Pharmaceuticals) was injected subcutaneously for analgesia.

### Experimental MI surgery
Mice were anesthetized with 4% isoflurane and subcutaneously injected with buprenorphine before being orally intubated and connected to a ventilator (SAR-830 small animal ventilator, GENEQ or Minivent 845, Hugo Sachs Harvard) that delivered a mixture of oxygen, room air and 2–3% isoflurane. The fur on the left side of the chest was removed, the paws were connected to electrocardiographic (ECG) sensors and the cardiac rhythm was monitored continuously. A thoracotomy was made between the fourth and the fifth rib, the pericardium was opened, and the left anterior descending artery was identified and ligated directly after the first bifurcation. The induction of MI was verified by characteristic changes on the ECG. In ischemia-reperfusion experiments, the suture was released after 45 min of ischemia. The ribcage was then closed with sutures, the isoflurane administration was stopped, and the mice were extubated when signs of spontaneous breathing were observed. After the procedure, the mice recovered in

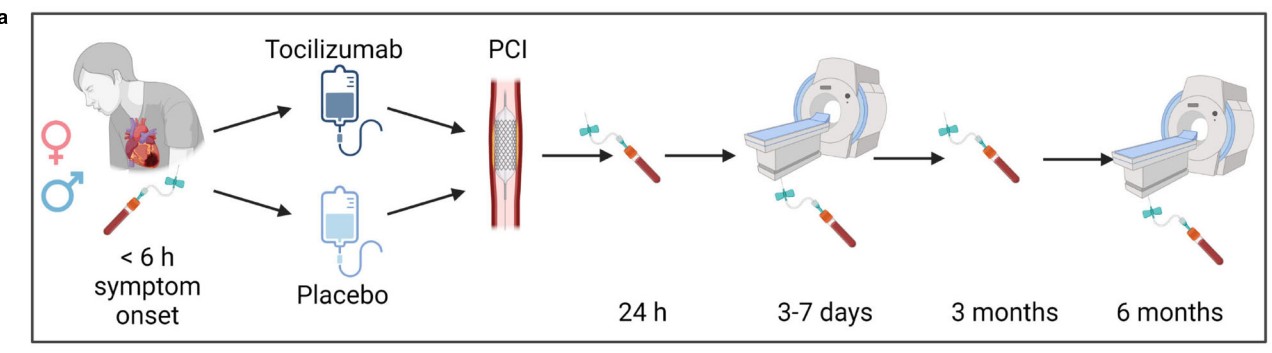

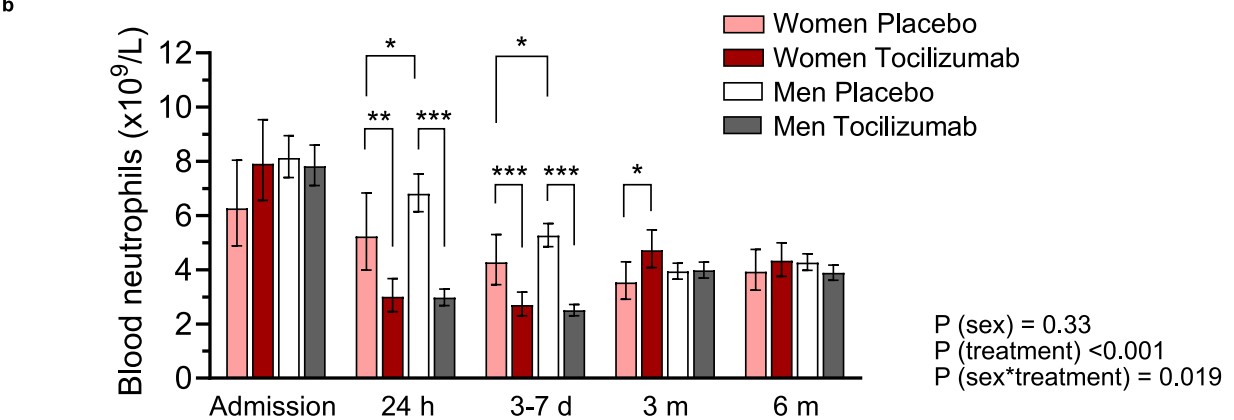

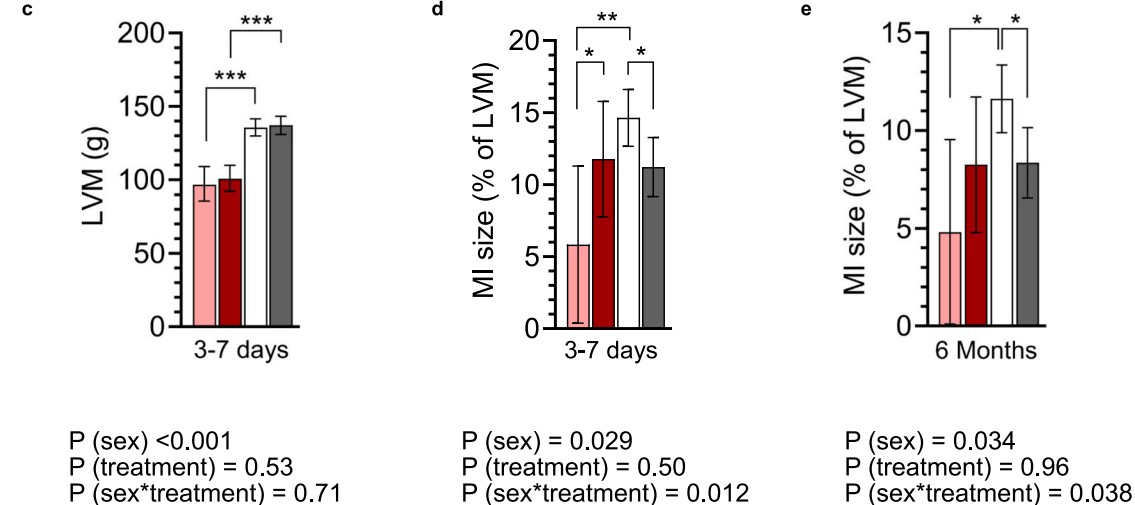

**Fig. 4 | Neutrophilia, cardiac injury, and response to anti-inflammatory treatment in acute MI are greater in men than women. a–e** Data from the double-blind ASSAIL-MI (ASSessing the effect of Anti-IL-6 treatment in MI) study. Patients (167 men, 32 women) with first-time ST-elevation MI were randomized to a single dose of the interleukin (IL)-6 receptor inhibitor tocilizumab or placebo prior to percutaneous coronary intervention (PCI). Men placebo; $n = 87$, men tocilizumab; $n = 80$, women placebo; $n = 11$ and women tocilizumab; $n = 21$. **a** Illustration of the study protocol. Created in BioRender. Dahl, T. (2024) https://BioRender.com/p48g864 **b** Blood neutrophil counts during the study. **c–e** Data from cardiac magnetic resonance imaging at 3–7 days (**c, d**) and 6 months (**e**) after randomization. LVM left ventricular mass. Bars indicate age-adjusted estimated marginal means, error bars are 95% confidence intervals (CI). Corresponding individual raw data are plotted in Supplementary Fig 9a (for Fig. 4b) and Supplementary Fig 11a–c (for Fig. 4c–e). *P* values are from two-sided repeated measures ANOVA (**b**) or ANCOVA (**c–e**) with LSD post-hoc tests. *$P < 0.05$, **$P < 0.01$, ***$P < 0.001$. Exact *p* values in post-hoc analyses (WP women placebo, MP men placebo, WT women tocilizumab, and MT men tocilizumab); neutrophil count in blood; 24 h; WP vs MP, $P = 0.018$, WP vs WT, $P = 0.002$ and MP vs MT, $P < 0.001$; 3–7 days; WP vs MP, $P = 0.025$, WP vs WT, $P < 0.001$ and MP vs MT, $P < 0.001$; 3 months; WP vs WT, $P = 0.019$. **c** LVM; 3–7 days; WP vs MP, $P < 0.001$ and WT vs MT, $P < 0.001$. **d** MI size; 3–7 days; WP vs MP, $P = 0.003$, WP vs WT, $P = 0.035$ and MP vs MT, $P = 0.012$. **e** MI size; 6 months; WP vs MP, $P = 0.016$, MP vs MT, $P = 0.012$.

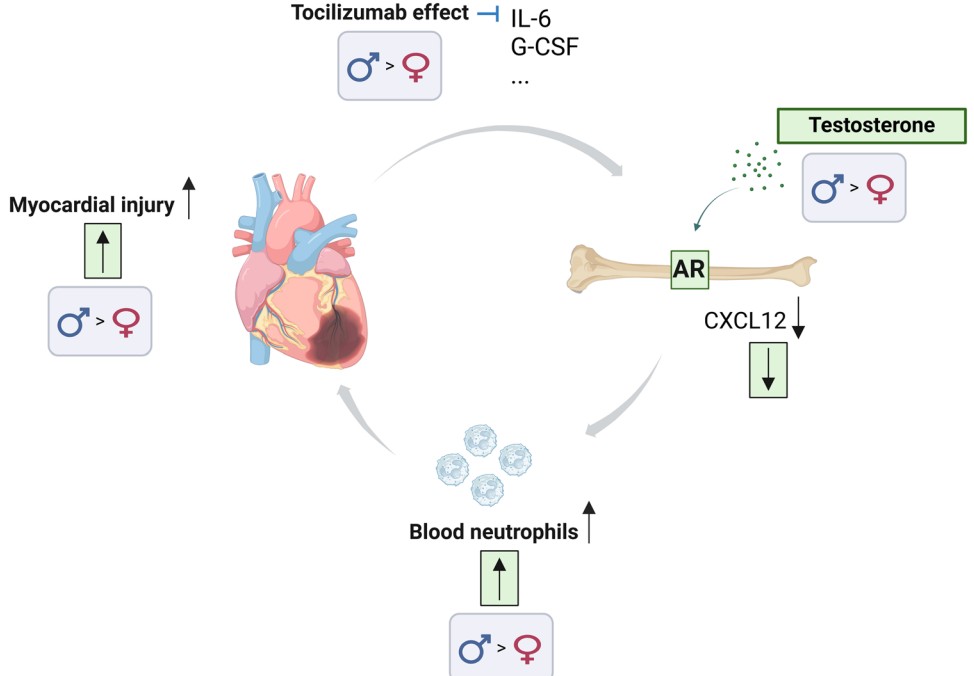

**Fig. 5 | Summary of findings and hypothesis.** In acute MI, the surge in blood neutrophils is a result of neutrophil release from the BM, triggered by various factors, including G-CSF, which suppress the production of neutrophil retention factors, most importantly CXCL12, in the BM. Testosterone (in green) acts via the AR to reduce the production of CXCL12 from BM stromal cells. Homeostatic lower levels of CXCL12 make the BM more prone to release neutrophils in response to egress triggers. The relatively higher blood neutrophils in the presence of male levels of testosterone aggravate myocardial injury in acute MI. Actions of testosterone in men translate into sex differences in neutrophilia and protection by tocilizumab in acute MI-reperfusion. Created in BioRender. Tivesten, Å. (2024) https://BioRender.com/b63s329. G-CSF granulocyte colony-stimulating factor, IL-6 interleukin-6, AR androgen receptor, BM bone marrow.

cages placed on heating pads. Sham-operated mice underwent the same procedure without tying the suture.

### Echocardiography in mice

To evaluate cardiac structure and function, a VEVO2100, VEVO3100, or VEVO770 echocardiography system (VisualSonics, Ontario, Canada) and the linear array transducers MS550D, MX250 or RMV704 were used. The systems included an integrated rail system for consistent positioning of the ultrasound probe. To minimize resistance to transmission of the ultrasonic beam, hair removal cream was applied to the chest before the examination. The mice were anesthetized with isoflurane (1.1%) and placed on a heating pad with their paws connected to ECG sensors. An optimal parasternal long-axis projection (i.e. visualization of the aortic and mitral valve and the maximum distance between the aortic valve and the cardiac apex) was acquired for orientation. The probe was rotated 90° and a parasternal short-axis cine loop of >1000 frames/s was acquired, with the ECG-gated kilohertz technique, at exactly 1, 3 and 5 mm below the mitral annulus. The stored data were evaluated offline (using Vevo® LAB desktop software version 5.5.0) in a blinded fashion. Infarct size/ area at risk was calculated from a 16-segment model based on three short-axis images. The myocardium at the basal and midventricular levels was divided into six segments, while the apical level was divided into four. Regional wall motion in each segment was scored 0 = normal, 1 = akinetic/dyskinetic full segment and 0.5 = for an akinetic/ dyskinetic part of a segment. The sum was then divided by 16 for percentage. Mice with infarct size/area at risk <30% or >60% of the LV at 24 h were excluded from the survival studies[74]. End-diastole was defined at the onset of the QRS complex, and end-systole was defined as the time of peak inward motion of the interventricular septum. Cardiac function parameters were calculated according to Simpson's calculations.

### Testosterone replacement in mice

Mice were bilaterally ORX or sham-ORX at 14-16 weeks of age. Following castration, mice were subcutaneously injected with vehicle (pure corn oil, catalog no. C8267, Sigma) or a physiological dose[75] of testosterone propionate (3 mg/kg/day, catalog no. 86541, Sigma) every 3 days for 3 weeks. A working solution of testosterone was made by dissolving 45 mg of testosterone propionate in 0.4 mL of 100% ethanol and then adding 30 mL of pure corn oil. MI was induced by permanent ligation of the left coronary artery and 24 h later, EDTA blood was collected.

### G-CSF-induced cell mobilization

Intraperitoneal injections of 125 μg/kg G-CSF (filgrastim, Neupogen®, Amgen) in 0.1% BSA in PBS or vehicle were given twice daily. In total, 5 injections were given. Mice were killed in the morning, 1 h after the last G-CSF injection, and EDTA blood for flow cytometry was collected.

### Assessment of infarct size

To evaluate infarct size after ischemia-reperfusion (45 min/24 h), mice were euthanized following 24 h reperfusion, by quickly excising the heart together with lungs and connecting tissue, which were immersed in 0.9% saline. Non-cardiac tissue was removed leaving the heart and aorta just proximal to the arch. The heart was perfused with 0.9% saline to remove any trapped blood/residual heme products. The previously occluded coronary artery was retied using the suture material that was left in situ to anatomically match the initial occlusion. Next, 5% (wt/vol) phthalocyanine blue (Sigma-Aldrich) in 0.9% saline was injected into the heart cavity through the ascending aorta to demarcate the ischemic area at risk (AAR) and remote area. The intact heart was then wrapped in clingfilm and stored at −20 °C for 24 h. While frozen, the heart was sliced transversely into 1 mm thick slices (n = 6-8/heart). The slices were incubated while continuously agitated with 1% (wt/vol)

2,3,5-triphenyltetrazolium chloride (TTC, Sigma-Aldrich) in 0.9% saline at 37 °C for 15 min to delineate the infarct area (IA). The slices were next placed in 10% neutral-buffered formalin for 30 min and then briefly immersed in PBS at room temperature. Using photos of the slices, the most central slice of the injured area was selected for quantification. The IA (pale), area at risk (AAR) (not blue) and total area of left ventricular (LV) myocardium were delineated manually and quantified using ImageJ 1.53t software (NIH, LOCI, University of Wisconsin, US) in a blinded manner.

## Tissue and blood collection

The mice were anesthetized (Isoflo vet, Orion Pharma Animal Health), blood was drawn from the LV, and the circulatory system was perfused with 0.9% saline (pH 7.4) under physiological pressure. Blood for smear, flow cytometry, total cell count (Sysmex KX-21, Sysmex Corporation) and plasma were collected in K3-EDTA Micro tubes (Sarstedt). Femur and tibia were dissected and cleaned from muscle tissue, epiphyses were removed, and BM cells harvested by flushing each bone with 5 mL of magnetic cell sorting (MACS) buffer (0.5% bovine serum and 2 mM EDTA in PBS, without $Ca^{2+}Mg^{2+}$). The cleaned and thoroughly flushed femur bones were snap frozen in liquid nitrogen and used for DNA analysis, and the femur and tibia BM was used for stromal cell stimulation ex vivo, flow cytometry or smears. Dissected femur and tibia were also used for sectioning of bone/BM. Spleen and heart were dissected and used for sectioning, flow cytometry and DNA analysis. Tissue for DNA analysis was kept at −80 °C, and tissue for flow cytometry, and sectioning was kept on ice, in PBS and 4% PFA, respectively.

## Magnetic cell sorting and preparation of cells for flow cytometry or RNA expression analysis

To obtain CD45⁻ (ORX and sham-ORX controls) or CD45⁻ TER119⁻ cells (O-ARKO and Cre⁺ controls), BM (ORX and sham-ORX controls) or bone shafts and BM (O-ARKO and Cre⁺ controls) were collected. Bones were flushed, and BM was collected. Bone shafts (O-ARKO and Cre controls) were cut into pieces in a small volume of digesting buffer: dispase II, 0.8 mg/mL (Life Technology); collagenase P, 0.2 mg/mL (Roche) and DNase I, 0.1 mg/mL (Worthington) in PBS (with $Ca^{2+}Mg^{2+}$). The BM cells were centrifuged at $300 \times g$ at +4 °C and the cell pellet, together with the bone pieces (for O-ARKO and Cre⁺ controls), was digested with gentle mixing (400 rpm) at 37 °C in 1 mL digestion buffer. After 15 min of incubation, the first fraction was harvested. Cells were gently resuspended, and the supernatant (fraction 1) was transferred to 5 mL of MACS buffer (0.5% BSA and 2 mM EDTA in PBS). Fractions were centrifuged and resuspended in 1 mL of MACS buffer. Fractions 2 and 3 were collected by repeating the steps above twice. Fractions 1–3 were pooled, centrifuged, and erythrocytes were hemolyzed in ammonium chloride buffer (0.16 M $NH_4Cl$, 0.13 M EDTA and 12 mM $NaHCO_3$ in $H_2O$). After two washes, the cells were resuspended in MACS buffer, filtered through a 70 μm cell strainer and counted with a NucleoCounter NC-100 (Chemimetec). Cells from one mouse (ORX and sham-ORX controls: 30-70 ×10⁶ cells from two femur bones) or two pooled mice (O-ARKO and Cre⁺ controls: Total 60-100 ×10⁶ cells from two tibia and femur bones) were CD45⁺ or CD45⁺ and TER119⁺ depleted according to the manufacturer´s instructions using anti-mouse CD45 or a 1 + 1 mixture of anti-mouse CD45 and TER119 conjugated MACS® MicroBeads: Miltenyi Biotec, 130-052-301 (clone 30F11.1) and 130-049-901 (clone TER119) respectively, and LD column (Miltenyi Biotec, 130-042-901). Cells from the negative fraction were collected, centrifuged, lyzed in RNeasy Plus Mini kit RLT buffer (Qiagen) and frozen at −80 °C prior to RNA preparation.

CD45⁺ leukocytes were isolated from the heart for flow cytometry. Briefly, the heart (LV and septum) was minced in a small volume of digestion buffer: collagenase I, 450 U/mL (Sigma); collagenase XI,

125 U/mL (Sigma); hyaluronidase I-S, 60 U/mL (Sigma) and DNase I, 60 U/mL (Worthington) and digested in 1 mL buffer with mixing (1400 rpm) for 1 h at 37 °C. The digested tissue was passed through a 70 μm cell strainer using a 5 mL syringe plunger. Cells were collected and washed in flow cytometry (FACS) buffer (2% FBS and 2 mM EDTA in PBS without $Mg^{2+}/Ca^{2+}$), stained and sorted according to the manufacturer´s instructions using anti-mouse CD45 conjugated MACS® MicroBeads (Miltenyi Biotec, 130-052-301, clone 30F11.1, 10 μL per 10⁷ cells) and LS columns (Miltenyi Biotec, #130-042-401, 10 μL per 10⁷ cells). The cells in the CD45 positive fraction were counted with a NucleoCounter NC-100 (Chemimetec).

To obtain single BM and spleen cells for flow cytometry, tissue was passed through a 70 μm cell strainer. To remove erythrocytes from BM, spleen, and blood, cells were lyzed for 5 min in ammonium chloride buffer, washed and resuspended in FACS buffer (2% heat-inactivated FCS and 2 mM EDTA in PBS), passed through a 70 μm cell strainer and counted in a Sysmex KX-21 (Sysmex Corporation).

## Flow cytometry

After Fc-blockage with anti-mouse CD16/CD32 (clone 2.4G2, BD Biosciences 553142, dil 1:100), expression of various cell-surface markers was detected using fluorochrome-conjugated antibodies (Supplementary Table 2). Neutrophils and monocytes in blood were gated using CD11b-APC, CD11b-PE-Cy7 (clone M1/70, Biolegend 101212 or 101216, dil 1:200) and Ly6G-PE-Cy7 (dil 1:100), Ly6G-AF488 (dil 1:100) or Ly6G-PE (dil 1:200) (Clone 1A8, Biolegend 127618, 127626 or 127608) or side scatter (SSC) together with Gr1-Fitc (clone RB6-8C5, eBioscience 11-5931, dil 1:100). B and T cells in blood were gated using CD11b-APC (clone M1/70, Biolegend 101212, 1:200), Ly6G-PE (clone 1A8, Biolegend 127618, dil 1:200), CD3-PE-Cy7 (clone 17A2, Biolegend 100220, dil 1:100) and CD19-Fitc (clone 1D3, BD Biosciences 553785, dil 1:100). Neutrophils in spleen were gated using SSC and Gr1-Fitc (clone RB6-8C5, eBioscience 11-5931, dil 1:100). Neutrophils in BM were gated using CD45-AF488 (clone 30F11, Biolegend 103105, dil 1:200), CD11b-APC (clone M1/70, Biolegend 101212, dil 1:200) and Ly6G-PE (clone 1A8, Biolegend 127618, dil 1:200). Common myeloid progenitors were gated using a Fitc labeled lineage cocktail (Biolegend 133302, 20 μL per 10⁶ cells), c-kit-APC (clone 2B8, eBioscience 17-1171, dil 1:200) and Sca-1-PE (clone D7, eBioscience 12-5981, dil 1:200)[29]. Neutrophils in heart (CD45 MACS bead sorted positive fraction) were gated using SSC and Gr1-Fitc (clone RB6-8C5, eBioscience 11-5931, dil 1:100). Monocytes and macrophages in heart (CD45 MACS bead sorted positive fraction) were gated using CD11b-PE (clone M1/70, Biolegend 101207, dil 1:100) and F4/80-APC (clone BM8, eBioscience 17-4801, dil 1:100). Cells were analyzed in an Accuri C6 or FACS Aria (both BD Biosciences) and FlowJo software (Tree Star, Ashland, OR, USA) was used for data analysis. Fluorochrome-minus-one (FMO) was used as a control in all flow cytometry experiments, and logical axes were used in all fluorescence plots.

## Sorting of tdTomato-positive and negative BMSCs

Stromal cells from 30-week-old male Osx1-Cre⁺ tdTomato⁺ BM were prepared as previously described[23] with minor modifications, by flushing femur and tibia bones with complete medium: αMEM (Gibco) supplemented with 10% fetal calf serum (FCS; Corning®, 35-079-CV) and 1% penicillin-streptomycin (HyClone™). Cells were seeded in complete medium in 6-well plates at $1.2 \times 10^6/cm^2$ and cultured at 37 °C and in humidified air containing 5% $CO_2$. After four days nonadherent cells were removed by changing medium. Medium was hereafter changed every three days. After 10 days cell supernatant was removed, cells were washed with PBS and adherent stromal cells were digested in dispase II (0.8 mg/mL, Life Technology), collagenase P (0.2 mg/mL, Roche) and DNase I (0.1 mg/mL, Worthington) in PBS (with $Ca^{2+}Mg^{2+}$) for 60 min at 37 °C. Cells were gently removed using a cell scraper,

washed in complete medium, resuspended in FACS buffer (2% heat-inactivated FCS and 2 mM EDTA in PBS) and (Osx1-Cre⁺) tdTomato⁺ and tdTomato⁻ cells were sorted on a FACS Aria Fusion (BD Biosciences) using the 561 nm laser. Osx1-Cre⁻ tdTomato⁺ cells were used as a negative control.

## Blood and BM smears
Blood smears were obtained from EDTA blood. BM smears were prepared by flushing the femur with 1 mL 2 mM EDTA in PBS. The cell suspension was centrifuged at $400 \times g$ for 3 min, and a smear was made from the cell pellet. The smears were fixed in 100% methanol for 2 min and stained with May-Grunewald Giemsa. The leukocyte differentials in blood and the myeloid cell morphology in BM was manually assessed by a blinded observer using light microscopy and an Axiophot 100× oil objective (Zeiss), and the differentials of each cell type (in blood: lymphocytes, neutrophils, eosinophils, and monocytes, and in BM: myeloblasts, promyelocytes, myelocytes, metamyelocytes, bands and segmented neutrophils) were calculated from a total of 200 blood cells and 500 BM cells.

## Stromal cell stimulation ex vivo
Stromal cells from male C57BL/6 J BM were prepared as previously described[23] with minor modifications, by flushing femur and tibia with complete medium: αMEM (Gibco) supplemented with 10% FCS (Corning®, 35-079-CV) and 1% penicillin-streptomycin (HyClone™). Cells were seeded complete medium in 6-well plates at $1.2 \times 10^6/cm^2$ and cultured at 37 °C and in humidified air containing 5% $CO_2$. After four days nonadherent cells were removed by changing medium and cells were stimulated in αMEM supplemented with 2% hormone-deprived (dextran charcoal-stripped) FCS (Gibco, A3382101), 1% PEST and the AR agonist dihydrotestosterone (DHT, Sigma A8380) or vehicle control. Medium was changed every three days and fresh DHT or vehicle was added. After 10 days cell supernatant was collected, centrifuged at $300 \times g$ for 5 min and frozen at −80 °C. The adherent stromal cells were washed with PBS and digested in dispase II (0.8 mg/mL, Life Technology), collagenase P (0.2 mg/mL, Roche) and DNase I (0.1 mg/mL, Worthington) in PBS (with $Ca^{2+}Mg^{2+}$) for 60 min at 37 °C. Cells were gently removed using a cell scraper, washed in complete hormone-stripped FCS medium, lyzed in RLT Plus buffer (Qiagen) and frozen at −80 °C.

## Ar DNA quantification
In the ARKO mouse model exon 2 of the Ar gene is excised. The presence of exon 2 versus exon 3 was therefore used to quantify the efficacy of the AR knockout. Genomic DNA from femur and spleen was isolated using the DNeasy Blood and Tissue kit (Qiagen). DNA amplification was analyzed in triplicate wells using Power SYBR Green Master Mix (Life Technology) in ViiA7 Real-Time PCR System (Applied Biosystems). The following primer pairs were used: Ar exon 2, forward GGACCATGTTTTACCCATCG and reverse CCACAAGTGAGAGCTCCGTA, and Ar exon 3, forward TCTATGTGCCAGCAGAAACG, and reverse CCCAGAGTCATCCCTGCTT. The level of Ar exon 2 was normalized to Ar exon 3 using the $2^{-\Delta\Delta ct}$ method.

## RNA isolation and real-time qPCR
Total RNA was extracted using RNeasy Plus Mini kit (Qiagen, cat.no. 74134) according to manufacturer's instructions. cDNA was synthesized using High-Capacity cDNA Reverse Transcription Kit (Applied Biosystems, cat.no. 4374967). RNA expression was analyzed in duplicate wells using TaqMan® Gene Expression Assays (Applied Biosystem, cat.no. 4331182) and TaqMan® Fast Advanced Master mix (Applied Biosystem, cat.no. 4444963) in ViiA7 Real-Time PCR System (Applied Biosystems). The TaqMan® Gene Expression Assays used were: Cxcl12 (Mm00445553_m1), 18 s (hs99999901_s1), Hprt1 (Mm00446968_m1). Data was normalized to the reference gene 18 s (CD45⁻ and

CD45⁻TER119⁻ cells) or Hprt1 (BMSCs) and gene expression was calculated using the $2^{-\Delta\Delta ct}$ method.

## Single-cell RNA sequence analysis
The normalized counts already processed[32] were downloaded (GSE128423; https://www.ncbi.nlm.nih.gov/geo/query/acc.cgi?acc=GSE128423) and analyzed in Seurat/4.1.0[76] within R/4.1.3 (R Core Team (2022). R: A language and environment for statistical computing. R Foundation for Statistical Computing, Vienna, Austria. URL https://www.R-project.org) as follows: the 2000 most variable features were used to generate the principal component analysis (PCA) of the RNA expression of Ar and Cxcl12. The top 10 components with the resolution of 0.86 were used to cluster the data onto Uniform manifold Approximation and Projection (UMAP) visualizations. To confirm that the resulting clusters 2 and 10 correspond to cluster 1 (leptin receptor-multipotent mesenchymal stem/stromal cells, Lepr-MSC) from the original analysis[32], the clusters were colored by lepr RNA expression. To confirm that the resulting cluster 11 corresponds to cluster 7 (osteolineage cells, OLC-1) from the original analysis[32], the cluster was colored by Bglap RNA expression.

## Sectioning and staining of bone, spleen, and heart tissue
Tissue was fixed in freshly prepared, ice-cold, 4% formaldehyde solution (Sigma-Aldrich, cat. no. 8.18708) with gentle mixing for 24 h at +4 °C. Bones were decalcified with 10% EDTA/PBS (Scharlau, cat. no. AC0965025P) solution at pH 8.1 and +4 °C by mixing gently for 3 days, with daily changes of the EDTA solution. Samples were transferred to a 30% sucrose (Sigma, cat. no. S9378) solution for 24 h before embedding in OCT medium (Tissue-Tek, cat. no. 4583). The tissue was sectioned at a 30 μm thickness and air-dried for 2 h at room temperature. Sections were re-hydrated with PBS for 10 min at room temperature, before adding nuclear stain Hoechst 33342 (Invitrogen, cat. no. H3570) for 5 min (1:1000 dilution in PBS). Sections were mounted with Fluoroshield (Sigma, cat. no. F6182) and imaged with Leica SP8 Laser confocal microscope and Nikon Spinning Disk confocal microscope.

## Myeloperoxidase ELISA
EDTA blood from mice was immediately centrifuged for 10 min at $2000 \times g$ and +4 °C. Plasma was frozen at −80 °C for later analysis. Myeloperoxidase was measured using an MPO Immunoassay (Hycult Biotech, HK210).

## Cardiac troponin I
EDTA blood from mice was immediately centrifuged for 10 min at $2000 \times g$ and +4 °C. Plasma was frozen at −80 °C for later analysis. Troponin I was measured using the Abbott high-sensitive assay on Alinity (Abbott Laboratories) with a limit of quantification of 2 ng/L and a CV < 10% within the range measured.

## CXCL12 ELISA
The concentration of CXCL12 protein in supernatant from BMSCs from DHT-treated male C57BL/6 J mice was analyzed using a Mouse CXCL12/SDF-1α Immunoassay (R&D Systems, MCX120). The CXCL12 concentration was normalized to the total protein concentration using the BCA Protein Assay Kit (Pierce™). Data below the limit of detection (LOD) were set to LOD/√2.

## Serum testosterone
Serum levels of testosterone were measured by an in-house validated gas chromatography-tandem mass spectrometry assay[77]. For testosterone, accuracy over two spiking levels is 105-114%, lower LOD 4 pg/mL, lower limit of quantification (LLOQ) 8 pg/mL, intra- and inter-assay coefficients of variations 1.6% and 2.8%, respectively[77]. All male mice and 7 out of 10 female mice had serum testosterone above LLOQ of the assay. Values below LLOQ were set to LLOQ/√2 in analyses[78].

## The ASSAIL-MI trial

In the phase 2 ASSAIL-MI (ASSessing the effect of Anti-IL-6 treatment in MI) trial (Clinicaltrials.gov: NCT03004703), we investigated the effect of a single dose of intravenous tocilizumab (280 mg) on improving myocardial salvage against placebo in patients admitted with acute STEMI. The full inclusion and exclusion criteria, as well as the study design, have been published elsewhere[18]. The trial protocol was approved by the regional ethics committee (Regional Committee for Medical Research Ethics South East Norway; 2016/1223-1) and all participants provided written informed consent. There was no participant compensation. An independent Data and Safety Monitoring Board oversaw the safety of the trial. The trial was approved by The Norwegian Medicines Agency and was conducted in compliance with the Declaration of Helsinki. Arterial blood samples were collected at admission, just before percutaneous coronary intervention, before intra-arterial unfractionated heparin and intravenous study medication were administered at the catheterization laboratory. Thereafter, venous blood samples were collected at 14–33 h (24 h), at 3–7 days, at 3 months, and at 6 months and leukocytes and differential counts were analyzed on Sysmex XN-10 (Sysmex, Kobe, Japan) per routine. High-sensitivity C-reactive protein (CRP) and N-terminal pro-brain natriuretic peptide (NT-proBNP) were analyzed on a MODULAR platform (Roche Diagnostics, Basel, Switzerland), and high-sensitivity troponin T (TnT) was measured by electrochemiluminescence immunoassay (Elecsys 2010 analyzer, Roche Diagnostics). Cardiac magnetic resonance imaging was performed on 1.5-T systems (Siemens Avanto, Philips Ingenia) using a gadolinium contrast agent as previously described[18]. All images were analyzed by a core laboratory at the Department of Circulation and Medical Imaging, Norwegian University of Science and Technology, Trondheim, Norway, using the Segment software (Medviso, Lund, Sweden)[79]. Infarct size was quantified using expectation maximization, weighted intensity, and a priori information method with manual correction. This method has been experimentally and clinically validated and agrees well with expert delineation[80]. Sub-studies on inflammation and stratification on sex were pre-specified in the originally approved study protocol. Baseline characteristics based on assigned sex and treatment of the study population are described in Supplemental Table 1.

## Statistical analysis

Statistical analyses were performed using Prism software (version 10.1.1, GraphPad software Inc). Data were analyzed for normality with or without transformation using descriptive statistics (skewness: −0.8 to 0.8 was accepted) and D´Agostino-Pearson test. When $n < 7$, normality of residuals was tested by D´Agostino-Pearson Omnibus (K2) test. Data that passed normality test was analyzed by two-sided Student $t$ test. Variances were tested for inequality by $F$ test and in case of unequal variances Student $t$ test with Welch´s correction was used. Non-normally distributed data were analyzed by Mann-Whitney $U$ test. Survival was analyzed by log-rank test and frequencies by Fisher's exact test.

Statistical analyses of ASSAIL-MI trial data were performed using SPSS version 29.0.0.0. Baseline characteristics stratified by sex and treatment were analyzed using one-way ANOVA with Bonferroni as post-hoc test or Kruskal-Wallis with Mann-Whitney as post-hoc test, depending on sample distribution. Categorical variables were tested using Chi-square. Leukocyte counts in ASSAIL-MI trial data that were skewed were $\log_{10}$-transformed before statistical analysis. Analyses of sex differences within treatment groups were performed by repeated measures ANOVA with sex as between-subject factor and age as covariate. In addition, treatment effects within sex groups were analyzed in a similar way. Post-hoc analyses were performed by general linear model analysis at each time-point with neutrophils as the dependent variable, sex or treatment group as fixed factors, and age as covariate. Cardiac magnetic resonance imaging measures were analyzed by ANCOVA with the measure as dependent, sex and treatment as fixed factors and age as covariate with LSD post-hoc tests.

## Reporting summary

Further information on research design is available in the Nature Portfolio Reporting Summary linked to this article.

## Data availability

All mouse data generated or analyzed during this study are included in this published article (and its supplementary information files). Ethical restrictions from the Regional Committee for Medical and Research Ethics in Southeast Norway prohibit data from individual patients from being made available on publicly available repositories. However, an institutional data transfer agreement can be established, and data can be shared if the aims of data use are covered by ethical approval and patient consent. The procedure will involve an update to the ethical approval as well as a review by legal departments at both institutions, and the process will typically take 2–4 months from initial contact. Access requests may be sent to Kaspar Broch (sbbrok@ous-hf.no). The annotated mouse single-cell gene expression data to evaluate *Ar* and *Cxcl12* expression in stromal cells were retrieved from GEO (GSE128423; https://www.ncbi.nlm.nih.gov/geo/query/acc.cgi?acc=GSE128423). Source data are provided with this paper.

## Code availability

The R code used for analysis of mouse single cell gene expression data has been deposited to a GitHub repository (DOI:10.5281/zenodo.14221831)[81].

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

## Acknowledgements

The authors thank Azra Miljanovic, Annelie Carlsson, Chauk Kabbouch, and Andreas Landin (Wallenberg Laboratory for Cardiovascular and Metabolic Research, University of Gothenburg) for excellent technical assistance, Rosie Perkins, University of Gothenburg, for editing the manuscript, and Sanna Abrahamsson from the Bioinformatics and Data Center at the Sahlgrenska Academy and Clinical Genomics Gothenburg at SciLifeLab for bioinformatics analyses and support. This work was supported by the Swedish Research Council (grant 2021-01856 to Å.T. and 2020-02298 to A.S.C.), the Swedish Heart-Lung Foundation (grant 20210492 to Å.T. and 20210588 to O.H.), the Novo-Nordisk Foundation (grant NNF18OC0034464 to Å.T. and NNF21OC0070314 to A.S.C.), grants from the Swedish state under the agreement between the Swedish government and the county councils (the ALF-agreement; ALFGBG-277435 to Å.T., ALFGBG-966178 to A.S.C. and 73190 to O.H.) and the Norwegian Health Authority SouthEast (grant 2019067 to B.H.). The ASSAIL-MI main study was supported by an independent grant from ROCHE who also provided drugs/placebo for infusion.

## Author contributions

E.S.E., M.L.R., B.H., I.J., A.K.F.M., P.A., K.B., L.G., B.H.A., G.Ø.A., M.H.T., and Å.T. conceived and designed the project. E.S.E., M.L.R., B.H., I.J., A.K.F.M., A.S.W., C.H., T.U., K.B., L.G., B.H.A., G.Ø.A., M.H.T., A.C., D.T., O.H., B.R., J.B., T.B.D., and Å.T. acquired the data. E.S.E., M.L.R., B.H., I.J., A.K.F.M., A.S.W., C.H., T.U., B.H.A., M.C.I.K., A.C., O.H., B.R., A.S.C., T.B.D., and Å.T. analyzed the data. E.S.E., M.L.R., B.H., I.J., A.K.F.M., C.H., P.A., K.B., L.G., G.Ø.A., M.C.I.K., M.H.T., A.C., O.H., B.R., J.B., E.O., M.C.L., A.S.C., T.B.D., and Å.T. interpreted the data. E.S.E., I.J., A.K.F.M., and Å.T. drafted the manuscript. E.S.E., M.L.R., B.H., I.J., C.H., T.U., P.A., K.B., M.C.I.K., B.R., J.B., M.C.L., A.S.C., T.B.D., and Å.T. revised the manuscript. All authors approved the final version of the manuscript.

## Funding

## Competing interests
K.B. has received lecture fees from Amgen, AstraZeneca, Boehringer, Novartis, NovoNordisk, Pharmacosmos, Pfizer, and consultant fees from AstraZeneca, Boehringer, Pharmacosmos, and Pfizer. L.G. has received lecture fees from AstraZeneca, Boehringer Ingelheim, Novartis, and Amgen. The remaining authors declare no competing interests.

## Additional information

ns licence, unless indicated otherwise in a credit line to the material. If material is not included in the article's Creative Commons licence and your intended use is not permitted by statutory regulation or exceeds the permitted use, you will need to obtain permission directly from the copyright holder. To view a copy of this licence, visit http://creativecommons.org/licenses/by/4.0/.

© The Author(s) 2025

[1]Wallenberg Laboratory for Cardiovascular and Metabolic Research, Department of Molecular and Clinical Medicine, Institute of Medicine, Sahlgrenska Academy at University of Gothenburg, Gothenburg, Sweden. [2]Research Institute of Internal Medicine, Oslo University Hospital Rikshospitalet, Oslo, Norway. [3]Faculty of Medicine, Institute of Clinical Medicine, University of Oslo, Oslo, Norway. [4]The Finsen Laboratory, Copenhagen University Hospital - Rigshospitalet, Copenhagen, Denmark. [5]Biotech Research and Innovation Centre (BRIC), Faculty of Health Sciences, University of Copenhagen, Copenhagen, Denmark. [6]Department of Medicine, Cardiovascular Division, Brigham and Women's Hospital, Harvard Medical School, Boston, MA, USA. [7]Thrombosis Research Center (TREC), Division of Internal Medicine, University Hospital of North Norway, Tromsø, Norway. [8]Department of Cardiology, Oslo University Hospital Rikshospitalet, Oslo, Norway. [9]K. G. Jebsen Cardiac Research Centre and Centre for Heart Failure Research, University of Oslo, Oslo, Norway. [10]Clinic of Cardiology, St. Olav's Hospital, Trondheim University Hospital, Trondheim, Norway. [11]Department of Circulation and Medical Imaging, Norwegian University of Science and Technology (NTNU), Trondheim, Norway. [12]Department of Cardiology, Oslo University Hospital Ullevål, Oslo, Norway. [13]Department of Microbiology, Tumor, and Cell Biology, Karolinska Institute, Karolinska University Hospital, Stockholm, Sweden. [14]Department of Internal Medicine and Clinical Nutrition, Institute of Medicine, Sahlgrenska Osteoporosis Centre, Centre for Bone and Arthritis Research at the Sahlgrenska Academy, University of Gothenburg, Gothenburg, Sweden. [15]Department of Rheumatology and Inflammation Research, Institute of Medicine, Sahlgrenska Academy at University of Gothenburg, Gothenburg, Sweden. [16]Department of Clinical Immunology and Transfusion Medicine, Region Västra Götaland, Sahlgrenska University Hospital, Gothenburg, Sweden. [17]Department of Physiology and Pharmacology, Karolinska Institute, Stockholm, Sweden. [18]Department of Laboratory Medicine, Institute of Biomedicine, University of Gothenburg, Gothenburg, Sweden. [19]Department of Endocrinology, Sahlgrenska University Hospital, Region Västra Götaland, Gothenburg, Sweden. [20]These authors contributed equally: Marta Lantero Rodriguez, Bente Halvorsen. ✉e-mail: asa.tivesten@medic.gu.se

