## [Transparent Peer Review file · Nature Communications]

Testosterone exacerbates neutrophilia and cardiac injury in myocardial infarction via actions in bone marrow

Corresponding Author: Professor Åsa Tivesten

Version 0:

Reviewer comments:

Reviewer #1

(Remarks to the Author)

In this manuscript, Eriksson et al. employed mouse models and human clinical data to explore the influence of testosterone on sex-specific cardiac injury following myocardial infarction (MI), with particular emphasis on neutrophils. Previous literature has shown that males have a disadvantage in acute MI outcomes and that neutrophils are involved in myocardial injury post-MI. This has been shown through cardiac infiltration of neutrophils and circulating neutrophil counts correlating positively with infarct size, ventricular arrhythmias, development of heart failure and mortality. To delve deeper into the sex differences observed in myocardial infarction (MI), particularly concerning neutrophils, the research team conducted initial experiments revealing that male mice exhibit heightened neutrophilia following MI compared to females. Furthermore, they found that this elevation in neutrophil levels is driven by testosterone. Next, the authors sought to uncover which cells were responding to the testosterone to induce the neutrophilia. Using an Osterix (Osx)-Cre mouse model, the androgen receptor (AR) was deleted from bone osteoprogenitors and bone marrow stromal cells (BMSCs). Interestingly, this experiment provided evidence that androgen signaling on BMSCs is what drives neutrophilia in male mice following MI (mimicking the results from castrating male mice). Following this observation, the group was interested in the molecular mechanism underlying this observation. They found that androgen signaling in BMSCs reduced the expression of CXCL12 which is involved in the retention of neutrophils in the bone marrow (both in vivo and in vitro). The group next turned their attention to human clinical trial data involving Anti-IL-6 treatment immediately following acute ST-elevation MI (STEMI). They found that this treatment significantly reduced circulating neutrophil levels in both men and women but was more exaggerated in men. In addition, the therapy was shown to decrease MI size more markedly in men than in women. Taken together, this manuscript has shown that testosterone, via the AR in BMSCs, exacerbates the egress of neutrophils from the BM in acute MI and worsens post-MI injury in males.

Although this study provides meaningful contributions to explain the mechanism for increased neutrophilia after MI in males, there are significant shortcomings including use of relatively young mice to model a medical condition that predominantly occurs in elderly individuals, lack of n for women included in the ASSAIL-MI trial and the disconnect between mouse and human data presented.

Major Comments

1. Although addressed as a limitation in the discussion, due to the extremely small number of women involved in the ASSAIL-MI trial the language and conclusions in the results section regarding the human data are likely overstated and should be adjusted accordingly. (the low number of women participants reduces the power to say anything about what is happening in the women participants). Also, comments should be included about the human data involving almost exclusively white individuals at the very least in the discussion section.
2. Because the medical condition being studied (MI) predominantly occurs in elderly individuals, justification should be provided for using young adult mice (3-5 months) instead of appropriately aged mice. Were any studies conducted on aged mice? If so, are the results consistent with young mice or are there key differences? If not, the relatively young age of mice should be explicitly discussed as a caveat in the discussion section.
3. Currently there is no direct connection between the mouse and human data (mouse focuses on CXCL12 whereas human focuses on IL-6). If there any known connection between IL-6 and CXCL12 involving neutrophil egress from the bone marrow? Were any experiments conducted to modulate the IL-6 signaling axis in mice?
4. No testosterone measurements are provided in either the mouse or human data, since this study implicates testosterone in sex-dimorphic MI outcomes, measurements should be provided. If this could not be done, the authors should explain why.
5. The scripts used for the reanalysis of the single cell RNA-seq could not be located. These should be provided as a

supplementary file or as a github repository for reproducibility and transparency purposes. A text description of analysis is insufficient for publication.

Minor Comments

1. For Figure 3f-l instead of just providing cluster identity as a number, it would be more useful to include the annotated cell type for each cluster.
2. Cxcl12 should not be capitalized when referring to the mouse gene/protein as per international nomenclature standards.
3. If possible, place individual dots per patient for Figure 4b,c like done in mouse data figures, so dispersion and variability between individuals is clear.

Reviewer #2

(Remarks to the Author)

Synopsis

It is known that men develop larger myocardial infarct size than women. In this manuscript, Eriksson et al report in this well-crafted manuscript that the blood neutrophil counts post MI were higher in male than in female mice. Reducing testosterone levels or genetic disruption of androgen receptor in bone marrow progenitor cells attenuated the increase in neutrophil count and myocardial infiltration by neutrophils, cardiac injury, and LV dilatation post MI. Testosterone worsens cardiac injury in acute MI and that castration of male mice protects against neutrophilia, cardiac neutrophil infiltration, mortality, and cardiac dilatation after an MI. DHT downregulates the expression of CXCL12 in bone marrow stem cells through androgen receptor contributing to the increased release of neutrophils from the bone marrow after an MI. The authors also performed a post-hoc analyses of data from a randomized ASSAIL-MI trial and showed that men show greater neutrophilia than women after a first-time STEMI and that tocilizumab reduces neutrophil count and myocardial infarct size more in men than in women.

Significance and novelty of the findings in the context of the extant literature.

The findings of these studies are highly significant and have clinical implications. They provide a mechanistic explanation for the known sex differences in the size of the myocardial infarct size between men and women and between male and female mice. Furthermore, the post-hoc analyses of the ASSAIL-MI trial by the authors suggest that men and women may respond differently to anti-inflammatory therapies after an acute MI and support the authors' assertion that sex differences should be considered in evaluating the efficacy of anti-inflammatory therapies to reduce myocardial injury. The paper is well written and the data are generally described with great clarity.

The data show that castration and knockdown of androgen receptor in male mice reduce cardiac neutrophil accumulation, mortality and LV dilatation. However, the changes in circulating neutrophil count and neutrophil infiltration of the myocardium occur so rapidly that it is likely that these effects result from increased egress of neutrophils from the bone marrow.

Although the data generally support the authors' assertions and conclusions, as noted below, some aspects of the data are perplexing and need further clarification/ discussion in the paper.

Specific Comments

1. The castration model with and without T replacement differs from the human condition In that serum testosterone levels typically decline after a myocardial infarction in men while in the mouse castration model, serum testosterone levels are maintained at a supraphysiologic level by the SC injection of testosterone propionate. Did the authors measure serum testosterone levels in the mouse model or in the human studies after an acute myocardial infarction to determine whether serum testosterone levels decrease or increase after a myocardial injury and whether tocilizumab administration is associated with an increase or decrease in circulating testosterone levels? If the testosterone levels in men decrease after a myocardial infarction, how do the authors explain the increased neutrophil egress from the bone marrow and neutrophilia after an acute event? This does not negate the authors' important findings but may suggest additional complexity; is it possible that the sex differences in the effects of testosterone are at least partly epigenetic (during critical developmental windows) rather than activational?

2. It would be helpful to report serum testosterone levels in the male and female mice after an experimentally-induced myocardial infarction.

3. Discussion (Page 8, 4th paragraph). Several statements in this paragraph are not entirely accurate. Importantly, a large cardiovascular safety trial of testosterone replacement therapy in middle-aged and older men with testosterone deficiency found no significant differences in the incidence of MACE (or myocardial infarction or death due to cardiovascular causes) between the testosterone and placebo-treated men (PMID: 37326322). The authors should discuss the findings of this cardiovascular safety trial of testosterone replacement therapy in the context of their own findings from the mouse and human studies.

Also, the statement about the effect of GnRH agonist on testosterone levels is not quite accurate. The initial increase in serum testosterone levels is short-lived - typically 2 to 3 weeks and most of the cardiometabolic effects of testosterone after institution of androgen deprivation therapy are attributed mostly to the severe testosterone deficiency.

Additionally, the effects of testosterone on atherogenesis in animal models are highly variable. In humans, testosterone treatment has not been associated with acceleration of atherogenesis assessed using MDCT calcium scores or CCA-IMT (PMID: 26262795).

4. It was not clear whether androgen receptor knockdown, castration, and testosterone administration in castrated mice had any effect on other blood cell types in circulation and progenitors in the bone marrow. Those data should be reported. In mice, testosterone replacement therapy increases the numbers of myeloid series progenitors and the circulating counts of neutrophils, monocytes, platelets and red cells (cells derived from myeloid series progenitors).

5. Similar effects of testosterone treatment on counts of neutrophils, monocytes, platelets and red cells have been reported in hypogonadal men (PMID: 32485095). It is possible that the rapid effects of testosterone during an acute myocardial infarction are on neutrophil egress rather than on the bone marrow progenitors.

6. Figure 4. Only the data on neutrophil counts are shown. The data on other cells derived from myeloid series progenitors (monocytes, platelets, red blood cells) should also be reported and would be useful in understanding if the effects are selective to neutrophils suggesting predominant effect on neutrophil egress or on the bone marrow progenitors in which case, all the other cell types derived from the myeloid series progenitors might be affected.

7. ASSAIL-MI Trial and Figure 4. What were the troponin levels in men and women in response to tocilizumab? It would be good to add those in Figure 4 as additional markers of myocardial injury.

8. Additional clarifying questions:

Figure 1a. What was the exact time for blood draw for neutrophil count? 45 minutes or 24 hours after ligation?

Figure 1b and k. Why medians, and not means? If medians were used, should quartile range rather than SEM be shown

Reviewer #3

(Remarks to the Author)

The manuscript by Eriksson and colleagues investigated the roles of testosterone on neutrophilia following cardiac ischemia-reperfusion injury. This is a well-designed study with novel mechanistic insights. There are several issues to be clarified/added to improve and strengthen the authors' hypothesis.

Page 4, Lines 1-5. Please report the infarct size in addition to the area at risk between genders reported here. Did male mice have greater infarct size than female mice?

Page 4, lines 8-11. Please report the infarct size and area at risk between ORX and sham mice reported here. Did ORX mice have smaller infarct size than sham mice?

Page 4, lines 22-26. Why did the authors use a permanent ligation instead of I/R model? Could different model affect the results reported here? Can the results from post-MI compared to those from cardiac I/R? Similar model should have been used for this purpose.

Several studies already demonstrated that testosterone deprivation could result in a larger infarct size, arrhythmias and poor LV function (PMID:25822979, PMID:30930218, PMID:27543302, PMID:29844497), the authors need to put their insights into those findings compared to theirs. Despite those controversial findings, they might suggest that testosterone replacement could play different roles under testosterone-deprived condition vs physiological conditions vs aging.

Fig 2f. The infarct size of these 2 groups is needed. Please add them in the figure.

Version 1:

Reviewer comments:

Reviewer #1

(Remarks to the Author)

The authors addressed my previous concerns satisfactorily.

Reviewer #2

(Remarks to the Author)

This is the first revision of a manuscript that investigated the role of testosterone in the known sex differences in the size of the myocardial infarct size between male and female mice. The authors have been responsive to the previous comments and have included additional data that have strengthened this well written manuscript.

I have a few additional comments:

1. The authors mention that serum testosterone level was measured using a sensitive assay; however, the assay is not described. Some essential characteristics of the testosterone assay (LLOQ, intra- and inter-assay CV, specificity, etc) should be included.
2. Splenic hematopoiesis plays an important role in the pathogenesis of diseases including myocardial infarction. The data presented in the manuscript suggests that testosterone is promoting the egress of neutrophils. In this regard, the spleen could be particularly important because it acts as a reservoir of myeloid cells, which can be quickly egress from the spleen in response to acute stress such as that of an acute myocardial infarction. Although the splenic hematopoiesis was not investigated in this study, the authors should acknowledge the important role of spleen in acute stress response in the Discussion.
3. Supplementary Figure 4 shows that Ar was NOT knocked out in the spleen; how do the authors explain this as spleen is an additional site of hematopoiesis and a reservoir of myeloid progenitors in the mouse?

Reviewer #3

(Remarks to the Author)

The authors have satisfactorily responded to all queries.

Point by point response

Line references refer to pdf file "Manuscript_tracked changes"

AUTHOR REPLY #1-2: Response to editorial comments in separate Cover letter

REVIEWER COMMENTS

Reviewer #1 (Remarks to the Author):

In this manuscript, Eriksson et al. employed mouse models and human clinical data to explore the influence of testosterone on sex-specific cardiac injury following myocardial infarction (MI), with particular emphasis on neutrophils. Previous literature has shown that males have a disadvantage in acute MI outcomes and that neutrophils are involved in myocardial injury post-MI. This has been shown through cardiac infiltration of neutrophils and circulating neutrophil counts correlating positively with infarct size, ventricular arrhythmias, development of heart failure and mortality. To delve deeper into the sex differences observed in myocardial infarction (MI), particularly concerning neutrophils, the research team conducted initial experiments revealing that male mice exhibit heightened neutrophilia following MI compared to females. Furthermore, they found that this elevation in neutrophil levels is driven by testosterone. Next, the authors sought to uncover which cells were responding to the testosterone to induce the neutrophilia. Using an Osterix (Osx)-Cre mouse model, the androgen receptor (AR) was deleted from bone osteoprogenitors and bone marrow stromal cells (BMSCs). Interestingly, this experiment provided evidence that androgen signaling on BMSCs is what drives neutrophilia in male mice following MI (mimicking the results from castrating male mice). Following this observation, the group was interested in the molecular mechanism underlying this observation. They found that androgen signaling in BMSCs reduced the expression of CXCL12 which is involved in the retention of neutrophils in the bone marrow (both in vivo and in vitro). The group next turned their attention to human clinical trial data involving Anti-IL-6 treatment immediately following acute ST-elevation MI (STEMI). They found that this treatment significantly reduced circulating neutrophil levels in both men and women but was more exaggerated in men. In addition, the therapy was shown to decrease MI size more markedly in men than in women. Taken together, this manuscript has shown that testosterone, via the AR in BMSCs, exacerbates the egress of neutrophils from the BM in acute MI and worsens post-MI injury in males.

Although this study provides meaningful contributions to explain the mechanism for increased neutrophilia after MI in males, there are significant shortcomings including use of relatively young mice to model a medical condition that predominantly occurs in elderly individuals, lack of n for women included in the ASSAIL-MI trial and the disconnect between mouse and human data presented.

AUTHOR REPLY #3:

We are very grateful to the reviewer for the assessment of our work and valuable suggestions how to improve the study. For the issues raised above, please see our replies below.

Major Comments

1. Although addressed as a limitation in the discussion, due to the extremely small number of women involved in the ASSAIL-MI trial the language and conclusions in the results section regarding the human data are likely overstated and should be adjusted accordingly. (the low number of women participants reduces the power to say anything about what is happening in the women participants). Also, comments

should be included about the human data involving almost exclusively white individuals at the very least in the discussion section.

AUTHOR REPLY #4:

The following sentence has been revised such that the words “but not women” have been removed (Results, line 309): “Tocilizumab significantly reduced infarct size at 3-7 days in men ~~but not women~~ and there was a statistical interaction between sex and treatment on early infarct size (Fig 4d).”

The following sentence has been added to the Discussion (line 510): “.....and the high proportion of white participants limits the generalizability to other racial/ethnic groups.”

2. Because the medical condition being studied (MI) predominantly occurs in elderly individuals, justification should be provided for using young adult mice (3-5 months) instead of appropriately aged mice. Were any studies conducted on aged mice? If so, are the results consistent with young mice or are there key differences? If not, the relatively young age of mice should be explicitly discussed as a caveat in the discussion section.

AUTHOR REPLY #5:

We thank the reviewer for this important remark. MI has been reported in adult mice ranging in age from 6 wk to 36 months, but at 3–6 mo of age, mice are homogenous in physiological maturation, including body weight¹. We agree that studies of aging mice would increase the translational relevance of our study, but our experience indicates that MI surgery is more challenging in older mice (>20 weeks) causing critical loss of mice during experiments in our hands. We apologize that we cannot satisfy the reviewer’s request due to technical limitations.

The following words have been added to the limitations section of the Discussion (line 501): “We studied mice at 3–5 months of age, when they are homogenous in physiological maturation and body weight¹, but we did not determine effects in older mice.”

3. Currently there is no direct connection between the mouse and human data (mouse focuses on CXCL12 whereas human focuses on IL-6). If there any known connection between IL-6 and CXCL12 involving neutrophil egress from the bone marrow? Were any experiments conducted to modulate the IL-6 signaling axis in mice?

AUTHOR REPLY #6:

We have not addressed this question experimentally. The following text has been incorporated in the Discussion (line 436): “After MI, both circulating and myocardial levels of IL-6 increase, and cardiac fibroblasts are a major source of IL-6 following MI². A central role for IL-6 as a trigger of neutrophilia in human MI is supported by the strikingly lower blood neutrophil levels in the tocilizumab-treated group in ASSAIL-MI (Fig 5). In line with this notion, IL-6 has been shown to mobilize neutrophils from the BM into the circulating pool³, adding effects on BM neutrophil release to the list of potential cardioprotective actions of IL-6 receptor inhibition^{4,5}. While a reduction in BM CXCL12 has been implicated in the egress of BM neutrophils in acute MI⁶, a potential role of CXCL12 in the neutrophilic response specific to IL-6 remains unclear.”

To reflect the unclear connection between IL-6 and CXCL12, we have slightly modified the legend to Figure 5: (**Fig 5. Summary of findings and hypothesis.** In acute MI, the surge in blood neutrophils is a result of neutrophil release from the BM, triggered by various factors, including G-CSF and IL-6, which suppress production of neutrophil retention factors, most importantly CXCL12, in the BM.....)

4. No testosterone measurements are provided in either the mouse or human data, since this study implicates testosterone in sex-dimorphic MI outcomes, measurements should be provided. If this could not be done, the authors should explain why.

AUTHOR REPLY #7:

Data on serum testosterone levels in female and male mice as well as castrated and sham-castrated male mice has been added to the paper (Supplementary Fig 1a and 1c, respectively). Further, we have compared testosterone levels in male mice subjected to MI-surgery versus sham-surgery (Supplementary Fig 1b). In accordance, changes have been made in the Results (page 4) and Methods.

Further, the following text has been added to the Discussion (line 446): “Testosterone levels are more than 10-fold higher in men than in women⁷. Using a high-sensitivity assay, we showed that serum testosterone levels were 15-fold higher in male than female mice and observed an expected natural variation in testosterone levels between male mice⁸. We found no effect of MI (versus sham-MI) on testosterone levels in male mice 48 h after surgery. We did not measure testosterone levels in ASSAIL-MI and therefore cannot determine how testosterone levels were potentially affected by MI or tocilizumab in this trial. Testosterone levels in men have been reported to be lowered after acute MI (within 11 h)^{9,10}, although the reported reduction was small (-20%)¹⁰. Indeed, the complex interplay between acute/chronic inflammatory and stress conditions and anabolic hormones¹¹, including testosterone, may be highly relevant for the acute MI setting.”

5. The scripts used for the reanalysis of the single cell RNA-seq could not be located. These should be provided as a supplementary file or as a github repository for reproducibility and transparency purposes. A text description of analysis is insufficient for publication.

AUTHOR REPLY #8:

We apologize for this mistake. We have now made our code available (link in the manuscript under the heading CODE AVAILABILITY).

Minor Comments

1. For Figure 3f-I instead of just providing cluster identity as a number, it would be more useful to include the annotated cell type for each cluster.

AUTHOR REPLY #9:

We now show the identity (abbreviated names) of the clusters of interest in Fig 3f.

2. Cxcl12 should not be capitalized when referring to the mouse gene/protein as per international nomenclature standards.

AUTHOR REPLY #10:

We have now tried to follow the instructions from Nature: <https://www.nature.com/nrm/for-authors/preparing-your-submission> “Use standard gene and protein nomenclature — for example, human genes (uppercase and italic), human proteins (uppercase), mouse genes (first letter only uppercase and italic) and mouse proteins (uppercase).” If errors remain, we will make further changes as requested.

3. If possible, place individual dots per patient for Figure 4b,c like done in mouse data figures, so dispersion and variability between individuals is clear.

AUTHOR REPLY #11:

In addition to the age-adjusted estimated marginal means with 95% confidence intervals presented in Figure 4, box plots with individual dots per patient for raw data corresponding to Fig 4 b-e has been added to the manuscript (Supplementary Fig 9-11).

Reviewer #2 (Remarks to the Author):

Synopsis

It is known that men develop larger myocardial infarct size than women. In this manuscript, Eriksson et al report in this well-crafted manuscript that the blood neutrophil counts post MI were higher in male than in female mice. Reducing testosterone levels or genetic disruption of androgen receptor in bone marrow progenitor cells attenuated the increase in neutrophil count and myocardial infiltration by neutrophils, cardiac injury, and LV dilatation post MI. Testosterone worsens cardiac injury in acute MI and that castration of male mice protects against neutrophilia, cardiac neutrophil infiltration, mortality, and cardiac dilatation after an MI. DHT downregulates the expression of CXCL12 in bone marrow stem cells through androgen receptor contributing to the increased release of neutrophils from the bone marrow after an MI. The authors also performed a post-hoc analyses of data from a randomized ASSAIL-MI trial and showed that men show greater neutrophilia than women after a first-time STEMI and that tocilizumab reduces neutrophil count and myocardial infarct size more in men than in women.

Significance and novelty of the findings in the context of the extant literature.

The findings of these studies are highly significant and have clinical implications. They provide a mechanistic explanation for the known sex differences in the size of the myocardial infarct size between men and women and between male and female mice. Furthermore, the post-hoc analyses of the ASSAIL-MI trial by the authors suggest that men and women may respond differently to anti-inflammatory therapies after an acute MI and support the authors' assertion that sex differences should be considered in evaluating the efficacy of anti-inflammatory therapies to reduce myocardial injury. The paper is well written and the data are generally described with great clarity.

The data show that castration and knockdown of androgen receptor in male mice reduce cardiac neutrophil accumulation, mortality and LV dilatation. However, the changes in circulating neutrophil count and neutrophil infiltration of the myocardium occur so rapidly that it is likely that these effects result from increased egress of neutrophils from the bone marrow.

Although the data generally support the authors' assertions and conclusions, as noted below, some aspects of the data are perplexing and need further clarification/ discussion in the paper.

AUTHOR REPLY #12:

We are very grateful to the reviewer for the high appraisal of our work, nice words and useful suggestions how to further improve the study.

Specific Comments

1. The castration model with and without T replacement differs from the human condition In that serum testosterone levels typically decline after a myocardial infarction in men while in the mouse castration model, serum testosterone levels are maintained at a supraphysiologic level by the SC injection of testosterone propionate. Did the authors measure serum testosterone levels in the mouse model or in the human studies after an acute myocardial infarction to determine whether serum testosterone levels decrease or increase after a myocardial injury and whether tocilizumab administration is associated with an increase or decrease in circulating testosterone levels? If the testosterone levels in men decrease after a myocardial infarction, how do the authors explain the increased neutrophil egress from the bone

marrow and neutrophilia after an acute event? This does not negate the authors' important findings but may suggest additional complexity; is it possible that the sex differences in the effects of testosterone are at least partly epigenetic (during critical developmental windows) rather than activational?

AUTHOR REPLY #13:

We measured serum testosterone in the mouse model; we found no difference in serum testosterone 48h after MI/sham-MI surgery in male mice. These data have been added to the paper (Supplementary Fig 1b). In accordance, changes have been made in the Results (page 4) and Methods.

Further, the following text has been added to the Discussion (line 446): “Testosterone levels are more than 10-fold higher in men than in women⁷. Using a high-sensitivity assay, we showed that serum testosterone levels were 15-fold higher in male than female mice and observed an expected natural variation in testosterone levels between male mice⁸. We found no effect of MI (versus sham-MI) on testosterone levels in male mice 48 h after surgery. We did not measure testosterone levels in ASSAIL-MI and therefore cannot determine how testosterone levels were potentially affected by MI or tocilizumab in this trial. Testosterone levels in men have been reported to be lowered after acute MI (within 11 h)^{9,10}, although the reported reduction was small (-20%)¹⁰. Indeed, the complex interplay between acute/chronic inflammatory and stress conditions and anabolic hormones¹¹, including testosterone, may be highly relevant for the acute MI setting.”

2. It would be helpful to report serum testosterone levels in the male and female mice after an experimentally-induced myocardial infarction.

AUTHOR REPLY #14:

See AUTHOR REPLY #13.

3. Discussion (Page 8, 4th paragraph). Several statements in this paragraph are not entirely accurate. Importantly, a large cardiovascular safety trial of testosterone replacement therapy in middle-aged and older men with testosterone deficiency found no significant differences in the incidence of MACE (or myocardial infarction or death due to cardiovascular causes) between the testosterone and placebo-treated men (PMID: 37326322). The authors should discuss the findings of this cardiovascular safety trial of testosterone replacement therapy in the context of their own findings from the mouse and human studies.

AUTHOR REPLY #15:

We thank the reviewer for these valuable suggestions. We have added a new section in the Discussion regarding the cardiovascular effects of androgens in a broader perspective, including the reference PMID: 37326322 (Discussion, line 457):

“The clinical data on the cardiovascular actions of androgens are conflicting and there is a lack of understanding of underlying mechanisms. Some data link low testosterone levels and/or androgen deprivation to increased cardiovascular risk in men^{12,13}. Conversely, some evidence supports adverse effects of testosterone, including the positive association of genetically determined testosterone levels with risk of MI and heart failure in men¹⁴, and the increased cardiovascular risk in transmen on testosterone treatment¹⁵. Notably, a large trial of testosterone replacement therapy in men with testosterone deficiency found no significant differences in major adverse cardiac events, MI, or death due to cardiovascular causes between testosterone- and placebo-treated men¹⁶ and testosterone treatment for 3 years was not associated with acceleration of atherogenesis¹⁷. Among experimental studies, several report that castration/anti-androgen treatment of rodents protects against the adverse consequences of experimental MI¹⁸⁻²³, while others show no or beneficial effects of testosterone in

experimental MI models²⁴⁻²⁸. A potential explanation for these diverging results may be that different mechanisms are in play, including (anti)atherogenic²⁹, metabolic, direct cardiotropic³⁰ and immunomodulatory actions^{29,31}, and that their relative contribution varies depending on experimental settings, target tissues, cardiovascular diseases and disease stages.”

Also, the statement about the effect of GnRH agonist on testosterone levels is not quite accurate. The initial increase in serum testosterone levels is short-lived - typically 2 to 3 weeks and most of the cardiometabolic effects of testosterone after institution of androgen deprivation therapy are attributed mostly to the severe testosterone deficiency.

AUTHOR REPLY #16:

To avoid a too complex discussion, we have deleted the words “.....,the relatively poorer cardiovascular safety during the first year of androgen deprivation therapy using gonadotropin-releasing hormone agonists that promote an initial testosterone flare,.....” from the Discussion (page 10).

Additionally, the effects of testosterone on atherogenesis in animal models are highly variable. In humans, testosterone treatment has not been associated with acceleration of atherogenesis assessed using MDCT calcium scores or CCA-IMT (PMID: 26262795).

AUTHOR REPLY #17:

The above-mentioned reference has been cited (see AUTHOR REPLY #15 for details).

4. It was not clear whether androgen receptor knockdown, castration, and testosterone administration in castrated mice had any effect on other blood cell types in circulation and progenitors in the bone marrow. Those data should be reported. In mice, testosterone replacement therapy increases the numbers of myeloid series progenitors and the circulating counts of neutrophils, monocytes, platelets and red cells (cells derived from myeloid series progenitors).

AUTHOR REPLY #18:

We thank the reviewer for this comment. As O-ARKO completely mimics the effect of castration regarding neutrophil release, we chose to provide more in-depth data from the O-ARKO model. We have now further expanded the characterization of the O-ARKO model regarding other cell types and BM progenitors in the BM. The following data has been added to the manuscript:

- O-ARKO ischemia-reperfusion model: Besides neutrophils, we have added data on monocytes and lymphocytes as well as Ly6G-mean fluorescence intensity in neutrophils (Results and Supplementary Fig. 5).
- O-ARKO steady state: Besides the blood leukocyte differential count reported in the paper, we have added data on blood levels of monocytes, lymphocytes, platelets, and erythrocytes (Supplementary Fig 7). Further, besides the granulopoiesis reported in the paper, we have now added data on BM levels of neutrophils, Ly6G-mean fluorescence intensity in BM neutrophils and the number of BM common myeloid progenitors (Supplementary Fig 7).

We have also added the following text to the Discussion (line 406): “In addition to its effects on neutrophils, testosterone therapy dose-dependently increases the circulating counts of erythrocytes, monocytes, and platelets in men³² and androgen deprivation modestly lowers erythrocyte counts in men³³. These observations raise the question of whether androgen affects early myeloid progenitors in the BM. However, circulating counts of monocytes, platelets, and

erythrocytes and numbers of myeloid progenitors are not altered by whole-body AR knockout in male mice³⁴. Similarly, in our current study, O-ARKO male mice displayed no alterations in circulating counts of monocytes or erythrocytes, although blood platelet numbers were slightly reduced. Further, we did not detect any differences in the number of common myeloid progenitors in the BM of O-ARKO compared with control mice.”

5. Similar effects of testosterone treatment on counts of neutrophils, monocytes, platelets and red cells have been reported in hypogonadal men (PMID: 32485095). It is possible that the rapid effects of testosterone during an acute myocardial infarction are on neutrophil egress rather than on the bone marrow progenitors.

AUTHOR REPLY #19:

We have now expanded the Discussion (line 396): “BMSCs regulate both neutrophil egress and hematopoiesis^{35,36}. In the steady state, testosterone-deficient whole-body AR knockout or castrated male mice show a defective granulopoiesis from the later myelocyte stage and a reduced number of neutrophils in blood^{34,37-39}. In accordance, testosterone treatment of men dose-dependently increases blood neutrophil counts³². By contrast, we showed that steady state granulopoiesis and blood neutrophils were unaltered in O-ARKO mice. Our observation that neutrophilia after MI-reperfusion was lower both in castrated male mice and O-ARKO male mice compared with their respective controls suggests the importance of androgen/AR-mediated regulation of neutrophil egress, rather than granulopoiesis, in the acute MI setting.”

See also AUTHOR REPLY #18.

6. Figure 4. Only the data on neutrophil counts are shown. The data on other cells derived from myeloid series progenitors (monocytes, platelets, red blood cells) should also be reported and would be useful in understanding if the effects are selective to neutrophils suggesting predominant effect on neutrophil egress or on the bone marrow progenitors in which case, all the other cell types derived from the myeloid series progenitors might be affected.

AUTHOR REPLY #20:

We thank the reviewer for this comment. In addition to neutrophils, we have added data on monocytes, lymphocytes, and platelets from the ASSAIL-MI trial (Supplementary Fig 9). Red blood cells were not measured in the study.

Changes have also been made in the Results text (line 299): “In contrast to neutrophils, there were no statistically significant interactions between sex and treatment for other cell types examined, including monocytes, lymphocytes, and platelets (Supplementary Fig. 9b-d).”

7. ASSAIL-MI Trial and Figure 4. What were the troponin levels in men and women in response to tocilizumab? It would be good to add those in Figure 4 as additional markers of myocardial injury.

AUTHOR REPLY #21:

We agree and have added data on Troponin T levels from the ASSAIL-MI study (Supplementary Fig 10). Changes have also been made in the Results text (line 301): “Troponin T levels (analyzed at admission and 8 h, 16 h, 24 h, 3-7 days and 3 and 6 months after reperfusion) showed no significant interaction between sex and treatment; however, tocilizumab reduced troponin T levels in men at 16 h after reperfusion (Supplementary Fig. 10).

8. Additional clarifying questions:

Figure 1a. What was the exact time for blood draw for neutrophil count? 45 minutes or 24 hours after ligation?

AUTHOR REPLY #22:

We now clarify in Results (line 157): “We showed that blood neutrophil count at 24 h.....”

Figure 1b and k. Why medians, and not means? If medians were used, should quartile range rather than SEM be shown

AUTHOR REPLY #23:

Where medians were used (due to non-normally distributed data), we have now added inter-quartile range, instead of just showing individual data. Inter-quartile range has been added to Fig 1c (former Fig 1b), Fig 1 m (former Fig 1k), and Fig 3m.

Reviewer #3 (Remarks to the Author):

The manuscript by Eriksson and colleagues investigated the roles of testosterone on neutrophilia following cardiac ischemia-reperfusion injury. This is a well-designed study with novel mechanistic insights. There are several issues to be clarified/added to improve and strengthen the authors' hypothesis.

AUTHOR REPLY #24:

We thank the reviewer for the assessment and for the valuable comments, which we think have improved our study. For the issues raised, please see our replies below.

Page 4, Lines 1-5. Please report the infarct size in addition to the area at risk between genders reported here. Did male mice have greater infarct size than female mice?

AUTHOR REPLY #25:

We have added data on area at risk as well as infarct size in male and female mice to the paper (Fig 1b). Changes have also been made in the Results text (page 4), Methods and Discussion (the words "and increased infarct size" added on page 8).

Page 4, lines 8-11. Please report the infarct size and area at risk between ORX and sham mice reported here. Did ORX mice have smaller infarct size than sham mice?

AUTHOR REPLY #26:

We have added data on area at risk as well as infarct size in sham-castrated and castrated male mice to the paper (Fig 1d). Changes have also been made in the Results text (page 4), Methods and Discussion (the words "infarct size" added on page 8).

Page 4, lines 22-26. Why did the authors use a permanent ligation instead of I/R model? Could different model affect the results reported here? Can the results from post-MI compared to those from cardiac I/R? Similar model should have been used for this purpose.

AUTHOR REPLY #27:

We show data on castration and O-ARKO effects in male mice in both ischemia-reperfusion (revised Fig. 1c-d and Fig. 2b-c) and permanent ligation (revised Fig. 1e-l and Fig. 2d-k) models. Both reperfused and non-reperfused models are established experimental models that are regarded to provide complementary information; the reperfused model resembles the common clinical setting with reperfusion by percutaneous coronary intervention, while the non-reperfused model is suitable for studies of cardiac remodeling and rupture-related mortality¹. Neutrophil migration is a rapid response in both models. A testosterone replacement experiment would be relevant, and should ideally have been performed, in both models. We chose to perform the testosterone replacement in combination with permanent ligation because of the lower variability in our hands. This is now discussed as a limitation (Discussion, line 498): "While castration and O-ARKO effects in male mice were studied in both reperfused and non-reperfused MI, the effect of testosterone replacement was studied in the non-reperfused MI model only."

Several studies already demonstrated that testosterone deprivation could result in a larger infarct size, arrhythmias and poor LV function (PMID:25822979, PMID:30930218, PMID:27543302, PMID:29844497), the authors need to put their insights into those findings compared to theirs. Despite those controversial findings, they might suggest that testosterone replacement could play different roles under testosterone-deprived condition vs physiological conditions vs aging.

AUTHOR REPLY #28:

We thank the reviewer for these valuable suggestions. We have added a new section in the Discussion regarding the cardiovascular effects of androgens in a broader perspective, including the references mentioned above (Discussion, line 457):

“The clinical data on the cardiovascular actions of androgens are conflicting and there is a lack of understanding of underlying mechanisms. Some data link low testosterone levels and/or androgen deprivation to increased cardiovascular risk in men^{12,13}. Conversely, some evidence supports adverse effects of testosterone, including the positive association of genetically determined testosterone levels with risk of MI and heart failure in men¹⁴, and the increased cardiovascular risk in transmen on testosterone treatment¹⁵. Notably, a large trial of testosterone replacement therapy in men with testosterone deficiency found no significant differences in major adverse cardiac events, MI, or death due to cardiovascular causes between testosterone- and placebo-treated men¹⁶ and testosterone treatment for 3 years was not associated with acceleration of atherogenesis¹⁷. Among experimental studies, several report that castration/anti-androgen treatment of rodents protects against the adverse consequences of experimental MI¹⁸⁻²³, while others show no or beneficial effects of testosterone in experimental MI models²⁴⁻²⁸. A potential explanation for these diverging results may be that different mechanisms are in play, including (anti)atherogenic²⁹, metabolic, direct cardiotropic³⁰ and immunomodulatory actions^{29,31}, and that their relative contribution varies depending on experimental settings, target tissues, cardiovascular diseases and disease stages.”

Fig 2f. The infarct size of these 2 groups is needed. Please add them in the figure.

AUTHOR REPLY #29:

We have added data on area at risk as well as infarct size in O-ARKO and control mice to the paper (Fig 2c). Changes have also been made in the Results text (page 5), Methods and Discussion (page 8).

REFERENCES

1. Lindsey, M.L., *et al.* Guidelines for in vivo mouse models of myocardial infarction. *Am J Physiol Heart Circ Physiol* **321**, H1056-H1073 (2021).
2. Alter, C., *et al.* IL-6 in the infarcted heart is preferentially formed by fibroblasts and modulated by purinergic signaling. *J Clin Invest* **133**(2023).
3. Florentin, J., *et al.* Interleukin-6 mediates neutrophil mobilization from bone marrow in pulmonary hypertension. *Cell Mol Immunol* **18**, 374-384 (2021).
4. Nakao, T. & Libby, P. IL-6 helps weave the inflammatory web during acute coronary syndromes. *J Clin Invest* **133**(2023).
5. Broch, K., *et al.* Randomized Trial of Interleukin-6 Receptor Inhibition in Patients With Acute ST-Segment Elevation Myocardial Infarction. *J Am Coll Cardiol* **77**, 1845-1855 (2021).
6. Sreejit, G., *et al.* Neutrophil-Derived S100A8/A9 Amplify Granulopoiesis After Myocardial Infarction. *Circulation* **141**, 1080-1094 (2020).
7. Schiffer, L., *et al.* Classic and 11-oxygenated androgens in serum and saliva across adulthood: a cross-sectional study analyzing the impact of age, body mass index, and diurnal and menstrual cycle variation. *Eur J Endocrinol* **188**(2023).
8. Williamson, C.M., Lee, W., Romeo, R.D. & Curley, J.P. Social context-dependent relationships between mouse dominance rank and plasma hormone levels. *Physiol Behav* **171**, 110-119 (2017).
9. Geisthovel, W., Perschke, B., von zur Muhlen, A. & Klein, H. [Plasma testosterone, free testosterone fraction LH and FSH in males during the early stage of acute myocardial infarction (author's transl)]. *Z Kardiol* **68**, 776-783 (1979).
10. Pugh, P.J., Channer, K.S., Parry, H., Downes, T. & Jone, T.H. Bio-available testosterone levels fall acutely following myocardial infarction in men: association with fibrinolytic factors. *Endocr Res* **28**, 161-173 (2002).
11. Straub, R.H. Interaction of the endocrine system with inflammation: a function of energy and volume regulation. *Arthritis Res Ther* **16**, 203 (2014).
12. Ohlsson, C., *et al.* High serum testosterone is associated with reduced risk of cardiovascular events in elderly men. The MrOS (Osteoporotic Fractures in Men) study in Sweden. *J Am Coll Cardiol* **58**, 1674-1681 (2011).
13. Hu, J.R., *et al.* Cardiovascular Effects of Androgen Deprivation Therapy in Prostate Cancer: Contemporary Meta-Analyses. *Arterioscler Thromb Vasc Biol* **40**, e55-e64 (2020).
14. Luo, S., Au Yeung, S.L., Zhao, J.V., Burgess, S. & Schooling, C.M. Association of genetically predicted testosterone with thromboembolism, heart failure, and myocardial infarction: mendelian randomisation study in UK Biobank. *BMJ* **364**, l476 (2019).
15. Nota, N.M., *et al.* Occurrence of Acute Cardiovascular Events in Transgender Individuals Receiving Hormone Therapy. *Circulation* **139**, 1461-1462 (2019).
16. Lincoff, A.M., *et al.* Cardiovascular Safety of Testosterone-Replacement Therapy. *N Engl J Med* **389**, 107-117 (2023).
17. Basaria, S., *et al.* Effects of Testosterone Administration for 3 Years on Subclinical Atherosclerosis Progression in Older Men With Low or Low-Normal Testosterone Levels: A Randomized Clinical Trial. *JAMA* **314**, 570-581 (2015).
18. Cavašin, M.A., Sankey, S.S., Yu, A.L., Menon, S. & Yang, X.P. Estrogen and testosterone have opposing effects on chronic cardiac remodeling and function in mice with myocardial infarction. *Am J Physiol Heart Circ Physiol* **284**, H1560-1569 (2003).
19. Cavašin, M.A., Tao, Z., Menon, S. & Yang, X.P. Gender differences in cardiac function during early remodeling after acute myocardial infarction in mice. *Life Sci* **75**, 2181-2192 (2004).

20. Cavasin, M.A., Tao, Z.Y., Yu, A.L. & Yang, X.P. Testosterone enhances early cardiac remodeling after myocardial infarction, causing rupture and degrading cardiac function. *Am J Physiol Heart Circ Physiol* **290**, H2043-2050 (2006).
21. Froese, N., *et al.* Anti-androgenic therapy with finasteride improves cardiac function, attenuates remodeling and reverts pathologic gene-expression after myocardial infarction in mice. *J Mol Cell Cardiol* **122**, 114-124 (2018).
22. Hadi, N.R., Yusif, F.G., Yousif, M. & Jaen, K.K. Both castration and goserelin acetate ameliorate myocardial ischemia reperfusion injury and apoptosis in male rats. *ISRN Pharmacol* **2014**, 206951 (2014).
23. Wang, M., *et al.* Role of endogenous testosterone in myocardial proinflammatory and proapoptotic signaling after acute ischemia-reperfusion. *Am J Physiol Heart Circ Physiol* **288**, H221-226 (2005).
24. Tsang, S., Wu, S., Liu, J. & Wong, T.M. Testosterone protects rat hearts against ischaemic insults by enhancing the effects of alpha(1)-adrenoceptor stimulation. *Br J Pharmacol* **153**, 693-709 (2008).
25. Nahrendorf, M., *et al.* Effect of testosterone on post-myocardial infarction remodeling and function. *Cardiovasc Res* **57**, 370-378 (2003).
26. Pongkan, W., Chattipakorn, S.C. & Chattipakorn, N. Chronic testosterone replacement exerts cardioprotection against cardiac ischemia-reperfusion injury by attenuating mitochondrial dysfunction in testosterone-deprived rats. *PLoS One* **10**, e0122503 (2015).
27. Seara, F.A.C., *et al.* Paradoxical effect of testosterone supplementation therapy on cardiac ischemia/reperfusion injury in aged rats. *J Steroid Biochem Mol Biol* **191**, 105335 (2019).
28. Pongkan, W., *et al.* Vildagliptin reduces cardiac ischemic-reperfusion injury in obese orchietomized rats. *J Endocrinol* **231**, 81-95 (2016).
29. Wilhelmson, A.S., *et al.* Testosterone Protects Against Atherosclerosis in Male Mice by Targeting Thymic Epithelial Cells-Brief Report. *Arterioscler Thromb Vasc Biol* **38**, 1519-1527 (2018).
30. Marsh, J.D., *et al.* Androgen receptors mediate hypertrophy in cardiac myocytes. *Circulation* **98**, 256-261 (1998).
31. Wilhelmson, A.S., *et al.* Testosterone is an endogenous regulator of BAFF and splenic B cell number. *Nat Commun* **9**, 2067 (2018).
32. Gagliano-Juca, T., *et al.* Differential effects of testosterone on circulating neutrophils, monocytes, and platelets in men: Findings from two trials. *Andrology* **8**, 1324-1331 (2020).
33. Gagliano-Juca, T., *et al.* Mechanisms responsible for reduced erythropoiesis during androgen deprivation therapy in men with prostate cancer. *Am J Physiol Endocrinol Metab* **315**, E1185-E1193 (2018).
34. Chuang, K.H., *et al.* Neutropenia with impaired host defense against microbial infection in mice lacking androgen receptor. *The Journal of experimental medicine* **206**, 1181-1199 (2009).
35. Greenbaum, A., *et al.* CXCL12 in early mesenchymal progenitors is required for haematopoietic stem-cell maintenance. *Nature* **495**, 227-230 (2013).
36. Bruserud, O., Mosevoll, K.A., Bruserud, O., Reikvam, H. & Wendelbo, O. The Regulation of Neutrophil Migration in Patients with Sepsis: The Complexity of the Molecular Mechanisms and Their Modulation in Sepsis and the Heterogeneity of Sepsis Patients. *Cells* **12**(2023).
37. Wilhelmson, A.S., *et al.* Androgens regulate bone marrow B lymphopoiesis in male mice by targeting osteoblast-lineage cells. *Endocrinology* **156**, 1228-1236 (2015).
38. Li, F., *et al.* Sex differences orchestrated by androgens at single-cell resolution. *Nature* **629**, 193-200 (2024).
39. Markman, J.L., *et al.* Loss of testosterone impairs anti-tumor neutrophil function. *Nat Commun* **11**, 1613 (2020).

Point by point response

(Line references to pdf file "Manuscript_tracked changes")

REVIEWERS' COMMENTS

Reviewer #1 (Remarks to the Author):

The authors addressed my previous concerns satisfactorily.

We are grateful to the Reviewer for the assessment of our work.

Reviewer #2 (Remarks to the Author):

This is the first revision of a manuscript that investigated the role of testosterone in the known sex differences in the size of the myocardial infarct size between male and female mice. The authors have been responsive to the previous comments and have included additional data that have strengthened this well written manuscript.

We are grateful to the Reviewer for the assessment of our work.

I have a few additional comments:

1. The authors mention that serum testosterone level was measured using a sensitive assay; however, the assay is not described. Some essential characteristics of the testosterone assay (LLOQ, intra- and inter-assay CV, specificity, etc) should be included.

We have now expanded the description of the testosterone assay. The following text appears in Methods (line 844): "Serum levels of testosterone were measured by an in-house validated gas chromatography–tandem mass spectrometry assay¹. For testosterone, accuracy over two spiking levels is 105-114%, lower limit of detection 4 pg/ml, lower limit of quantification (LLOQ) 8 pg/ml, intra- and inter-assay coefficients of variations 1.6% and 2.8%, respectively¹. All male mice and 7 out of 10 female mice had serum testosterone above LLOQ of the assay. Values below LLOQ were set to LLOQ/√2 in analyses²."

2. Splenic hematopoiesis plays an important role in the pathogenesis of diseases including myocardial infarction. The data presented in the manuscript suggests that testosterone is promoting the egress of neutrophils. In this regard, the spleen could be particularly important because it acts as a reservoir of myeloid cells, which can be quickly egress from the spleen in response to acute stress such as that of an acute myocardial infarction. Although the splenic hematopoiesis was not investigated in this study, the authors should acknowledge the important role of spleen in acute stress response in the Discussion.

We thank the Reviewer for the insightful comment. As suggested, we have revised the Discussion to further acknowledge the role of the spleen in the response to acute stress. The revised paragraph (starting on line 494) is provided below:

“In response to acute stress such as MI, hematopoietic cells may egress from the spleen besides the BM.³ In our model, we found tomato-positive cells in the splenic stroma, in line with previous data that *Osx*-driven Cre expression is not entirely specific to bone⁴. However, we did not detect measurable levels of *Ar* DNA recombination in the spleen, likely reflecting low efficiency of *Osx*-driven Cre in splenic cells. Further, while the spleen plays a significant role in the inflammatory response to an MI,³ experimental evidence supports the role of the BM over spleen as a supplier of neutrophils in the early stages of an MI^{3,5}. Nevertheless, we cannot exclude a contribution of the AR in splenic stromal cells for the O-ARKO data reported here.”

3. Supplementary Figure 4 shows that *Ar* was NOT knocked out in the spleen; how do the authors explain this as spleen is an additional site of hematopoiesis and a reservoir of myeloid progenitors in the mouse?

Indeed, while *Osx*-Cre targets some cells in the spleen (Supplementary Fig. 4a), AR recombination within the spleen was not detectable (Supplementary Fig. 4b). This likely reflects the low efficiency of *Osx*-Cre activity in the spleen. This finding sharply contrasts with the results in bone (Supplementary Fig. 4b), highlighting that *Osx*-Cre is a much more effective tool for targeting stromal cells specifically in the skeleton, where it is widely utilized.

The low efficiency in the spleen may stem from two possible factors: either the fraction of cells in which AR was ablated is negligible compared to the total cell population in the spleen, or the level of Cre expression is low. While this level of expression is sufficient to induce recombination of the *Rosa*-tdTomato construct (given the high sensitivity of the Ai14 strain), it may not be adequate to induce recombination of AR.

This is incorporated into the Results section (line 209) as follows: “However, *Ar* DNA content, determined as the ratio between floxed exon 2 and intact exon 3, was reduced by 50% in the femur shaft but unaltered in the spleen and heart (Supplementary Fig. 4b), suggesting low efficiency of AR ablation in these tissues.” It is now also reflected in the Discussion (as specified in the response to comment #2 above).

Reviewer #3 (Remarks to the Author):

The authors have satisfactorily responded to all queries.

We are grateful to the Reviewer for the assessment of our work.

References

1. Nilsson, M.E., *et al.* Measurement of a Comprehensive Sex Steroid Profile in Rodent Serum by High-Sensitive Gas Chromatography-Tandem Mass Spectrometry. *Endocrinology* **156**, 2492-2502 (2015).
2. Handelsman, D.J. & Ly, L.P. An Accurate Substitution Method To Minimize Left Censoring Bias in Serum Steroid Measurements. *Endocrinology* **160**, 2395-2400 (2019).
3. Swirski, F.K., *et al.* Identification of splenic reservoir monocytes and their deployment to inflammatory sites. *Science* **325**, 612-616 (2009).
4. Chen, J., *et al.* *Osx-Cre* targets multiple cell types besides osteoblast lineage in postnatal mice. *PLoS One* **9**, e85161 (2014).
5. Sreejit, G., *et al.* Neutrophil-Derived S100A8/A9 Amplify Granulopoiesis After Myocardial Infarction. *Circulation* **141**, 1080-1094 (2020).